# Identification and characterization of a large family of superbinding bacterial SH2 domains

Tomonori Kaneko [1], Peter J. Stogios[2], Xiang Ruan[1], Courtney Voss[1], Elena Evdokimova[2], Tatiana Skarina[2], Amy Chung[3], Xiaoling Liu[1], Lei Li[1,4], Alexei Savchenko[2,5], Alexander W. Ensminger[3] & Shawn S.-C. Li[1,6]

Src homology 2 (SH2) domains play a critical role in signal transduction in mammalian cells by binding to phosphorylated Tyr (pTyr). Apart from a few isolated cases in viruses, no functional SH2 domain has been identified to date in prokaryotes. Here we identify 93 SH2 domains from *Legionella* that are distinct in sequence and specificity from mammalian SH2 domains. The bacterial SH2 domains are not only capable of binding proteins or peptides in a Tyr phosphorylation-dependent manner, some bind pTyr itself with micromolar affinities, a property not observed for mammalian SH2 domains. The *Legionella* SH2 domains feature the SH2 fold and a pTyr-binding pocket, but lack a specificity pocket found in a typical mammalian SH2 domain for recognition of sequences flanking the pTyr residue. Our work expands the boundary of phosphotyrosine signalling to prokaryotes, suggesting that some bacterial effector proteins have acquired pTyr-superbinding characteristics to facilitate bacterium-host interactions.

[1] Department of Biochemistry, Schulich School of Medicine and Dentistry, Western University, London, ON N6A 5C1, Canada. [2] Department of Chemical Engineering and Applied Chemistry, University of Toronto, Toronto, ON M5G 1L6, Canada. [3] Departments of Biochemistry and Molecular Genetics, MaRS Centre, West Tower, Floor 16, Toronto, ON M5G 1M1, Canada. [4] Cancer institute, the Affiliated Hospital of Qingdao University, Qingdao Cancer Institute and School of Basic Medicine, Qingdao University, 266021 Qingdao, China. [5] Department of Microbiology, Immunology and Infectious Diseases, University of Calgary, 2500 University Dr. NW, Calgary, AB T2N 1N4, Canada. [6] Children's Health Research Institute, Lawson Health Research Institute, 800 Commissioner's Road East, London, ON N6C 2V5, Canada. Correspondence and requests for materials should be addressed to S.S.-C.L. (email: sli@uwo.ca)

The bacterium *Legionella pneumophila* is the causative agent for a severe form of pneumonia called the Legionnaires' disease[1,2]. Since the discovery of *L. pneumophila* as a pathogenic bacterium infecting humans via the alveolar macrophage, at least 56 *Legionella* species have been identified, two-thirds of which are associated with human disease[2]. However, human-to-human transmission is exceedingly rare[3,4]. Instead, *Legionella* species exist as parasites of phylogenetically diverse protists in the natural environment[5,6]. To survive within the phagocytic host, the bacteria replicate inside a membrane-bound compartment called the *Legionella*-containing vacuole that incorporates materials hijacked from the endoplasmic reticulum and mitochondria of the host cell[7]. Numerous effector proteins are injected into the host cell upon *Legionella* infection[7,8]. For example, >330 effector proteins, accounting for ~10% of the *L. pneumophila* proteome, are injected into the host cell upon infection via the Icm/Dot type IV secretion system (T4SS)[6,9,10]. Many effector proteins contain eukaryotic motifs or domains such as the ubiquitin ligase U-box domain, the SET methyltransferase domain, and the protein kinase domain[10–13].

Protein tyrosine kinases play a critical role in regulating numerous cellular functions[14]. The tyrosine kinase signalling machinery comprises the tyrosine kinase (TK), the phosphotyrosine (pTyr) phosphatase (PTP), and the pTyr-binding module such as the Src homology 2 (SH2) or the phosphotyrosine-binding (PTB) domains. A recurring theme in pathogenic bacterium–host interactions involves host tyrosine kinases acting on bacterial effector proteins. For example, *Helicobacter pylori* and *Escherichia coli* can secrete effector proteins to function as substrates of the host tyrosine kinases. Once phosphorylated these effectors can hijack tyrosine kinase signalling in the host cell via the recruitment of host SH2 proteins[15,16]. *Salmonella*, *Yersinia*, and *Pseudomonas*, in contrast, secrete phosphotyrosine phosphatases to dephosphorylate host proteins[17–19]. Intriguingly, infection by *L. pneumophila* leads to tyrosine phosphorylation of proteins in both the host cell and the bacterium[20–22]. Regulation of tyrosine phosphorylation caused by *Legionella* infection was also reported for protozoan host species[23]. Tyrosine kinase or phosphotyrosine phosphatase inhibitors have been shown to reduce the uptake or replication of *L. pneumophila* in the host[21,24], suggesting an important role for the tyrosine kinase signalling machinery in the pathogen–host interaction and *Legionella* lifecycle inside the host.

Tyrosine phosphorylation may lead to changes in activity or subcellular localization of the substrate or the creation of binding sites for proteins containing an SH2 domain[25]. The human genome encodes >120 SH2 domains dedicated to the recognition of the pTyr, thereby ensuring that the kinase signal is transduced with high fidelity and efficiency[26]. Intriguingly, *Acanthamoeba castellanii*, a natural protozoan host for *Legionella*, harbors 48 putative SH2 domains, suggesting that tyrosine phosphorylation may play a role in the intracellular signalling of the protozoan host[27]. An SH2 domain typically contains ~100 amino acid residues and shares a common fold characterized by a seven-stranded β-sheet (strands βA to βG) flanked by two α-helices (αA and αB)[28]. Although different SH2 domains have distinct specificity, they all contain a pTyr-binding pocket and another pocket or binding site for the recognition of flanking residues, most often a C-terminal residue to the pTyr[29,30]. The critical role of the pTyr residue to SH2-binding is underscored in the observation that it contributes ~50% of the free energy of binding for a tyrosyl phosphopeptide to an SH2 domain[31,32]. Nevertheless, the affinity of an SH2 domain for its physiological ligand peptide is generally moderate, with the corresponding equilibrium dissociation constant in the 0.1–10 μM range[31,33].

The moderate affinity exhibited by the mammalian SH2 domains raised the question whether the SH2 fold can support greater affinities for the pTyr residue. We addressed this question by directed in vitro evolution of the pTyr-binding pocket in the Fyn SH2 domain using phage-displayed libraries. This led to the identification of an SH2 mutant, termed the SH2 superbinder, that bound to physiological pTyr peptides with affinities two to three orders of magnitude greater than the natural domain[34]. Because the SH2 superbinder, when introduced into mammalian cells, was capable of inhibiting tyrosine kinase signaling, we wondered if pathogenic bacteria would exploit this mechanism to gain an advantage when infecting a host. To explore this notion, we conducted an exhaustive search of the bacteria genome database and identified 93 putative SH2 domains in 84 *Legionella* proteins. We characterized 13 *Legionella* SH2 domains for ability to bind pTyr-containing peptides and found 11 were capable of binding to the pTyr residue and to mammalian proteins in a Tyr phosphorylation-dependent manner. Indeed, the affinities of some *Legionella* SH2 domains for the pTyr or phosphopeptides derived from mammalian proteins far exceeded those of a mammalian SH2 domain, suggesting that these bacterial SH2 domains are natural pTyr superbinders. Intriguingly, unlike the mammalian counterpart, a *Legionella* SH2 domain displayed no apparent sequence preference beyond the pTyr residue. Structural analysis of two *Legionella* SH2 domains revealed the basis for this unique mode of pTyr recognition. While both *Legionella* SH2 domains feature a defined pTyr-binding pocket, they are devoid of a second pocket or binding site for a C-terminal residue to the pTyr that is commonly found in a mammalian SH2 domain. Furthermore, we found that the majority (8/10) of the SH2-containing proteins were capable of translocating into human macrophage cells, suggesting that they may function as effector proteins. Our findings, which expand the realm of the SH2 domain from eukaryotes to prokaryotes, imply that the tyrosine kinase–pTyr–SH2 signaling axis may play an important role in the *Legionella*-host interaction and pathophysiology of the Legionnaires' disease.

## Results

**Legionella genomes encode numerous SH2-containing effectors.** A functional SH2 domain has not been identified in bacteria to date, though genomic analysis has predicted the existence of SH2 domain-containing effectors in *Legionella*[11,13]. Moreover, the Pfam domain database (version 31.0, March 2017) lists eight bacterial protein sequences with a putative SH2 domain[35]. We analyzed these putative SH2 domains by secondary structure prediction and found that they possessed the key characteristics of a typical SH2 domain (Fig. 1a). For example, LLO_2327, a protein from *L. longbeachae*, is predicted to form a β-sheet flanked by two α-helices found in a typical SH2 domain although it also contains a pancreatic polypeptide-like region not found in known SH2 domains. Importantly, it contains an Arg residue at the predicted βB5 position conserved in all functional SH2 domains characterized to date[34]. This finding prompted us to perform an exhaustive search of the bacterial genome sequences in the UniRef and NCBI databases. To minimize false identification, we employed secondary structure prediction to corroborate results from the sequence analysis (Fig. 1a). This structure-guided sequence analysis led to the identification of 93 putative SH2 domains from 84 proteins. The bacterial SH2 domains were clustered into ten groups based on sequence identity and domain organization (Figs. 1b and 2a; Supplementary Table 1). We named the different SH2 clusters LeSH (*Legionella* SH2), LeSH1a, LeSH1b, LeSH2, LeSH3, LeSH4, LeSH5, LUSH (*Legionella* U-box and SH2), RavO (Region allowing vacuole colocalization[9]), and

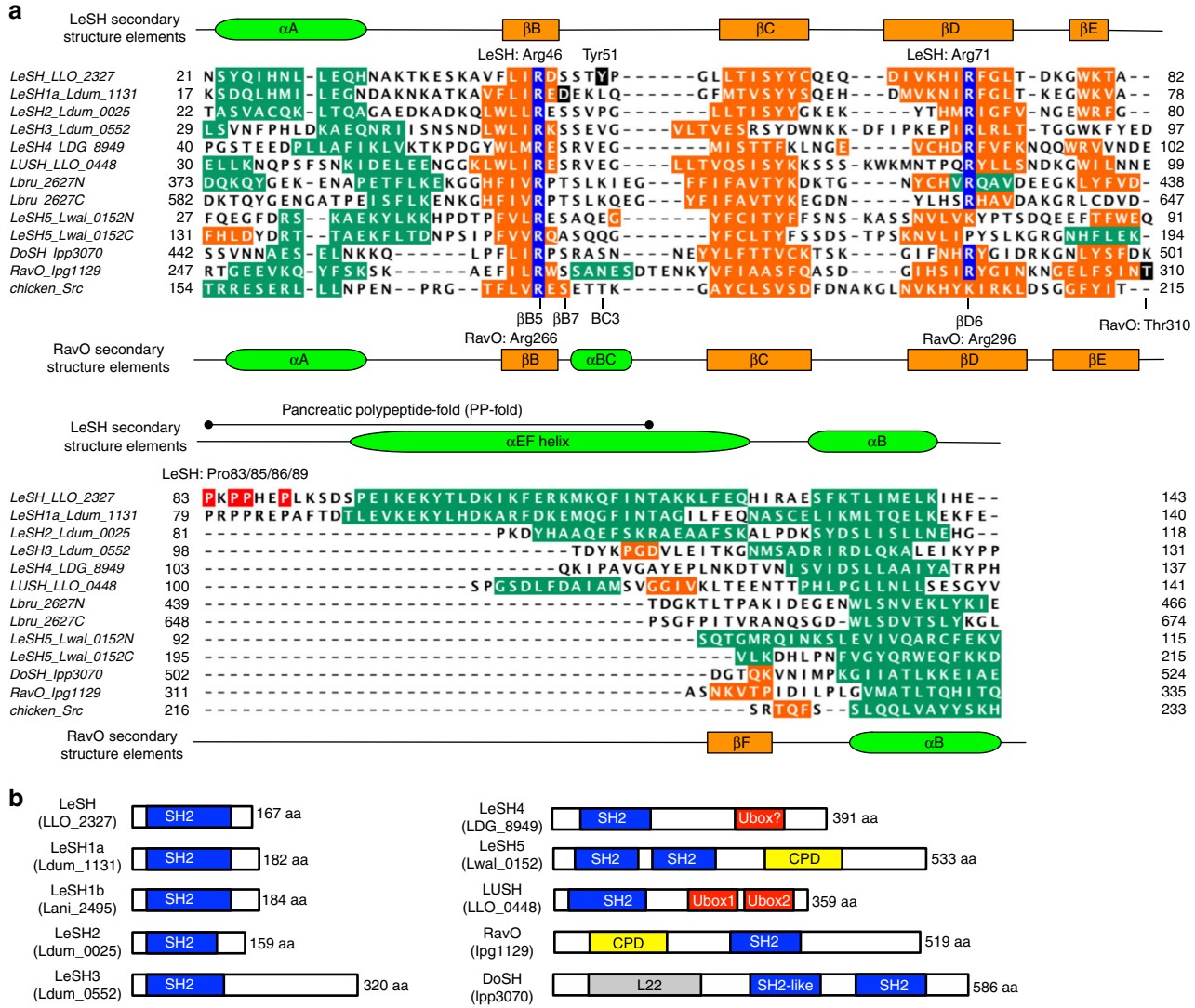

**Fig. 1** *Legionella* encodes a preponderance of SH2 domains. **a** Structure-guided sequence alignment of the *Legionella* SH2 domains. Shown are the SH2 domains from the *L. longbeachae* LeSH and LUSH; *L. dumoffii* LeSH1a, LeSH2, and LeSH3; *L. drancourtii* LeSH4; *L. pneumophila* RavO and DoSH (C-terminal SH2); and *L. waltersii* LeSH5 (Lwal_0152). The SH2 domain structures of LeSH, RavO, and Src were used as templates for the structure-guided alignment. Secondary structure elements are based on either solved (Src, LeSH, and RavO SH2 domains) or predicted structures. The α-helices are colored green and β-strands orange. The Arg residues at the βB5 and βD6 positions (based on secondary structure nomenclature of the Src SH2 domain[32]) are highlighted in blue. The four Pro residues in the pancreatic polypeptide-fold of LeSH are highlighted in red. **b** Domain organization of representative *Legionella* SH2 domain-containing proteins. CPD, cysteine protease domain[13]; L22, uncharacterized *Legionella* effector domain[13]

DoSH (Double SH2), respectively. The 93 candidate SH2 domains were derived from 40 *Legionella* species and two *Coxiellaceae* species (Fig. 2a) that are identified as bacterial symbionts of marine amoebae[36] and belong to the order *Legionellales*.

The bacterial SH2 sequences within a cluster generally align with each other in terms of secondary structure elements and the presence of the βB5-Arg residue[28] essential for pTyr recognition (Fig. 1a; Supplementary Figs. 1–3). The SH2 domains within the LeSH, LeSH1a or LeSH1b cluster are closely related to each other (>50% sequence identity, except for Lspi_0399), but are more distantly related to the LeSH2 SH2 domains due to the presence of a large sequence gap in the latter (Supplementary Figs. 1 and 2). LeSH, LeSH1a, and LeSH1b do not coexist with each other in a given *Legionella* species, whereas two or three paralogs of LeSH2 were identified in four *Legionella* species (Fig. 2a). In contrast, RavO was identified in five species including *L. pneumophila* (Supplementary Fig. 3a). Although the sequence identity of the SH2 domain between species varies from 16 to

44%, the βB5-Arg and flanking residues are conserved (Supplementary Fig. 3a and b). Intriguingly, except for *L. pneumophila*, the N-terminal DoSH SH2 domain from other species contains a His residue at the βB5 position (Supplementary Fig. 3a). Moreover, *L. cherrii* and *L. steigerwaltii* genomes each encodes two DoSH paralogs (Fig. 2a). Of the 10 *L. pneumophila* genome sequences that we surveyed, RavO was found in seven strains whereas DoSH in three (Supplementary Fig. 3c).

The presence of additional domains in some *Legionella* SH2-containing proteins suggests that they may be enzymes. For example, the LeSH5 and RavO clusters contain the bacterial cysteine protease domain (CPD)[13] that is found also in bacterial toxins[37,38]. Although the CPDs from the RavO and LeSH5 clusters showed 26–35% overall sequence identity to each other, they all contained the catalytic Cys residue (Fig. 1b; Supplementary Fig. 4a, b). The combination of a protease and an SH2 domain in the same protein is unique for RavO and LeSH5 since no known human proteases contain an SH2 domain.

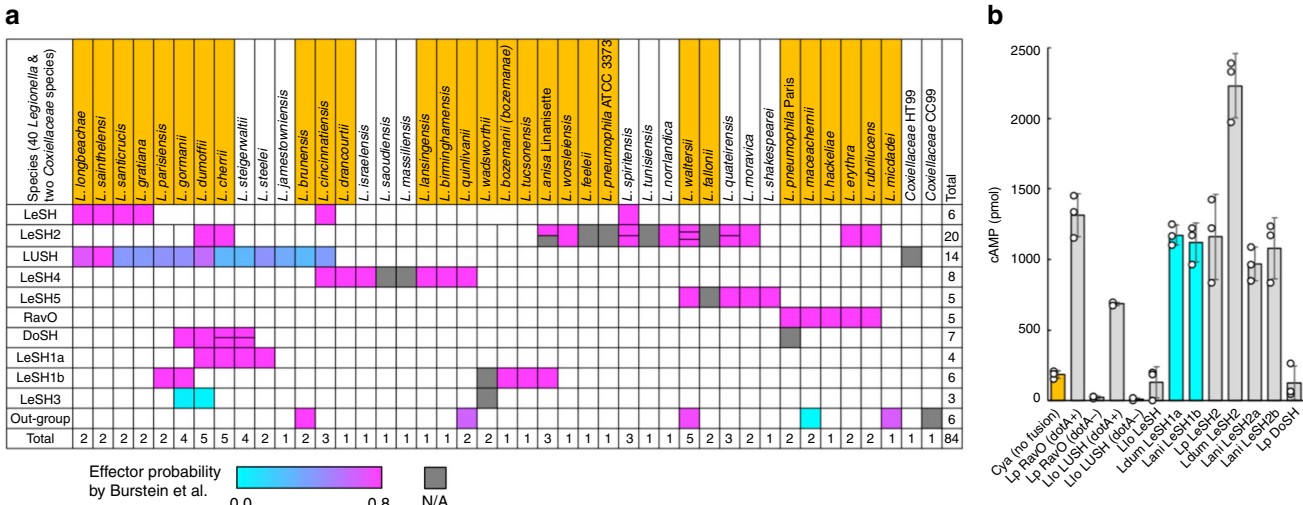

**Fig. 2** *Legionella* SH2 proteins are translocated effectors. **a** Distribution of SH2 domain proteins in different *Legionella* species and their effector probability. Included in the heat map are 40 *Legionella* species (including two strains from *L. pneumophila* that contain distinct SH2 proteins) and two species from the *Coxiellaceae* family (that belongs to the order *Legionellales*). The 28 *Legionella* species implicated in human disease are highlighted in yellow[2]. The effector prediction scores derived from Burstein et al.[13] were used to generate the heat map with a cyan-to-magenta gradient representing low-to-high effector probability for the 71 proteins listed in Burstein et al.[13]. The scores are not available (N/A) for 13 other SH2 proteins and they are colored gray. When LeSH2 or DoSH paralogs exist for a species, the cell is divided accordingly into two or three sections. For example, *L. waltersii* encodes three paralogs of LeSH2 all with high effector potential. See Supplementary Table 1 for a full list of the SH2 proteins. **b** Type IV secretion system (T4SS)-dependent translocation of *Legionella* SH2 proteins from bacteria to host cells. Ten SH2 proteins from *Legionella* fused with the adenylate cyclase Cya were expressed in the *L. pneumophila* strain Lp02 (dotA+). The *L. pneumophila* RavO and *L. longbeachae* LUSH were also tested in the Lp03 strain (defect in the T4SS, or dotA−). Differentiated U937 cells were infected with bacteria expressing the different Cya-fusion protein. Translocation of the fusion proteins was monitored by the intracellular cyclic AMP (cAMP) level, which is an indicator of the Cya activity in eukaryotic host cells. The error bars indicate standard deviation from triplicate infection experiments. See Supplementary Fig. 5 for expression level of the Cya-fusion proteins and *p*-values that indicate statistical significance of translocation of the Cya-fusion proteins in comparison to the Cya itself (labeled *no fusion*). The cAMP values are derived from Supplementary Fig. 5a, except for *L. dumoffii* LeSH1a and LeSH2, which are taken from Supplementary Fig. 5b. *L. dumoffii* LeSH1a and *L. anisa* LeSH1b, which are deficient in pTyr binding (see Fig. 3c), are colored cyan

In *L. pneumophila*, multiple ubiquitin ligases have been identified as effector proteins[39]. For example, the tandem U-box protein LubX (Supplementary Fig. 4c) has been found to target another *Legionella* effector protein SidH for ubiquitination after both are translocated into host cells[40,41]. The U-box is an E3 ubiquitin ligase domain structurally similar to the RING E3 ligase domain[42]. Intriguingly, the LUSH family proteins contain two U-boxes C-terminal to the SH2 domain[43] (Fig. 1b; Supplementary Fig. 4d; Supplementary Table 1). The domain organization in LUSH is similar to that in the Cbl family ubiquitin ligases that comprise an SH2 domain and a RING domain[44]. The mammalian Cbl ubiquitin ligase has been shown to catalyze the ubiquitination of tyrosine-phosphorylated proteins[44]. Consistent with this, we found that the *L. longbeachae* LUSH protein had E3 ligase activity in an in vitro ubiquitination assay employing the human ubiquitin-conjugating enzyme UBE2D2 (Supplementary Fig. 4e).

The SH2 domain-containing proteins are predicted effectors based on Burstein et al.[13], although the probability varies from one *Legionella* species to another (Fig. 2a; Supplementary Table 1). However, except for RavO, none of the SH2 proteins have been experimentally confirmed as effectors or substrates of the Dot/Icm T4SS system[9]. We determined the translocation efficiency for 10 SH2 proteins (including RavO used as a positive control) using the adenylate cyclase (Cya) reporter assay[8,45]. As expected, RavO was able to translocate to the U937 macrophages upon *L. pneumophila* infection in a Dot/Icm T4SS system-dependent manner (Fig. 2b). Importantly, seven of the nine predicted effectors were found to translocate efficiently to the host cell in this assay (Fig. 2b; Supplementary Fig. 5), suggesting the majority

of the SH2 containing proteins are secreted effectors. Nevertheless, translocation was not observed for the *L. longbeachae* LeSH and *L. pneumophila* DoSH. The failure in translocation for the *L. longbeachae* LeSH was unexpected since its homologs *L. dumoffii* LeSH1a (54% sequence identity; Supplementary Fig. 2) and *L. anisa* LeSH1b (55% identity) were able to translocate in the same assay. While it is possible that the *L. longbeachae* LeSH is not a secreted effector, it is more likely that a non-native effector may not be efficiently translocated in the *L. pneumophila* system employed here. Moreover, the efficient translocation of certain effectors may be facilitated by chaperones (e.g., the IcmSW complex[46]), and the *L. longbeachae* LeSH may require such a chaperone protein from the same species that may not be available in the *L. pneumophila* (strain Lp02) used in the Cya reporter assay.

**Legionella SH2 domains are bona fide pTyr superbinders**. To find out if the *Legionella* SH2 domains are functional pTyr binders, we determined their binding affinities for phosphotyrosine (pTyr) using the mini peptide GGpYGG that has been shown previously to bind engineered SH2 superbinders but not natural SH2 domains[39]. Intriguingly, the *L. longbeachae* LeSH, a 167-residue protein containing a single SH2 domain, bound the peptide with an equilibrium dissociation constant ($K_d$) of ~3.6 μM, but not to the Tyr- or pThr-containing version (Fig. 3a). To find out if other *Legionella* SH2 domains were also functional pTyr binders, we determined the affinities of 13 *Legionella* SH2 domains for the mini pTyr peptide. Compared to the wild-type Src SH2 domain that showed negligible binding ($K_d > 50$ μM),

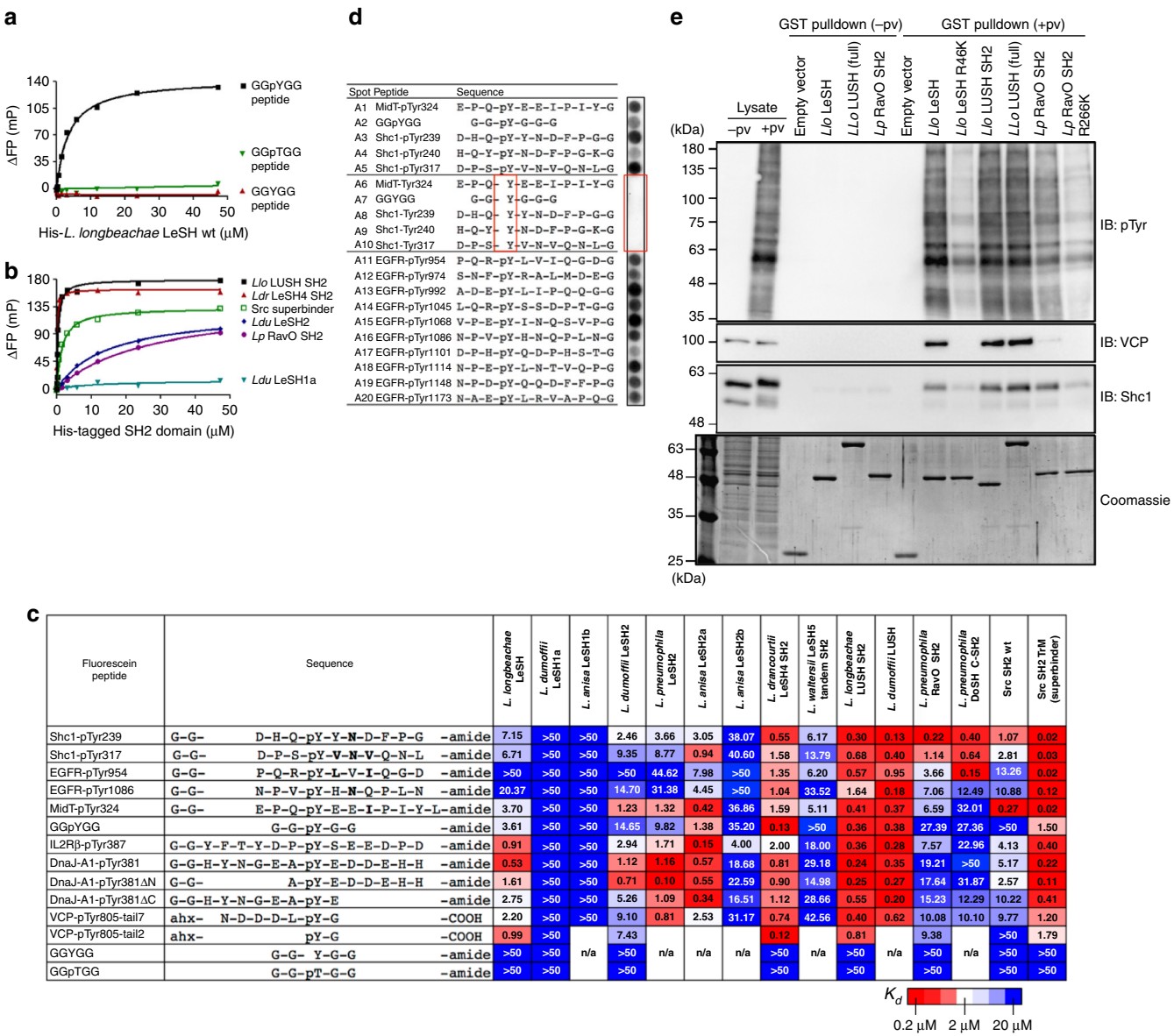

**Fig. 3** *Legionella* SH2 domains are bona fide phosphotyrosine superbinders. **a** Selective binding of *L. longbeachae* LeSH to the tyrosine-phosphorylated peptide GGpYGG, but not to the non-phosphorylated or pThr-containing versions. Peptides used for the fluorescence polarization assay were N-terminally labeled with fluorescein, through a GG- or a 6-aminohexanoic acid spacer. **b** Equilibrium binding curves (from fluorescence polarization assays) of five purified *Legionella* SH2 domains and the human Src SH2 triple mutant (Src superbinder) to the GGpYGG peptide. *Llo, L. longbeachae*; *Ldr, L. drancourtii*; *Ldu, L. dumoffii*; and *Lp, L. pneumophila*. **c** Equilibrium dissociation constants ($K_d$) of 13 *Legionella* SH2 domains for a selected group of pTyr, Tyr, or pThr-containing peptides. Except for the VCP peptide, the peptides used here were selected from the peptide arrays (Fig. 3d; Supplementary Fig. 6). The $K_d$ values are shown in μM, with a color gradient from red ($K_d = 0.2$ μM) to blue ($K_d = 20$ μM) to denote high to low affinity. See Supplementary Table 2 for fitting statistics of the binding curves. Ahx: 6-aminohexanoic acid spacer. **d** A peptide array probed with the GST-tagged RavO SH2 domain. Shown here is a strip of 20 spots representing pTyr peptides taken from the EGFR, Shc1 and MidT and the artificial peptide GGpYGG. Peptides A6–A10, identified in red rectangle, are non-phosphorylated versions of peptides A1–A5. See Supplementary Fig. 6a and b, for images of the full peptide arrays. **e** Binding of *Legionella* SH2 domains to tyrosine-phosphorylated proteins. Human macrophage-like U937 cells were treated with pervanadate (+pv) to enrich protein tyrosine phosphorylation before lysate is prepared. GST-fused *Legionella* SH2 domains were used to pull down the phosphorylated proteins from the lysate of U937-derived macrophage. Shown are western blotting images using antibodies specific for the pTyr, Shc1, or VCP. See Supplementary Fig. 6e for data on four additional *Legionella* proteins

6 of the 13 *Legionella* SH2 domains examined exhibited <10 μM affinities for the mini pTyr peptide (Fig. 3b, c; Supplementary Table 2 for curve fitting statistics). Furthermore, four of the bacterial SH2 domains bound the peptide more tightly than the Src SH2-derived superbinder. It is remarkable that the *L. drancourtii* LeSH4 SH2 domain bound to the GGpYGG peptide with 12-fold greater affinity than the superbinding Src SH2

mutant (Fig. 3c). These data suggest that some *Legionella* SH2 domains are natural pTyr superbinders. Of the three SH2 domains identified from *L. dumoffii*, the LUSH and LeSH2 SH2 domains bound the GGpYGG peptide with $K_d$ values of 0.38 and 14.65 μM, respectively, while the LeSH1a SH2 domain showed no binding (Fig. 3b, c). As shall become apparent later, LeSH1a contains a negatively charged Asp, instead of the usual Ser

residue, at the βB7 position of the pTyr-binding pocket, and the Asp potentially repels the pTyr due to an unfavorable electrostatic interaction. While it is not known why multiple SH2 domains with distinct affinities are present in the same species, the binding data suggest functional diversification among the SH2 effector paralogs.

To determine the specificity of the *Legionella* SH2 domains, we synthesized arrays of peptides representing physiological pTyr peptides that have been shown or are predicted by the SMALI program[47] to bind the Fyn or Grb2 SH2 domain. We then probed the peptide arrays with purified recombinant LeSH or RavO SH2 (Supplementary Fig. 6a and b). Both SH2 domains were able to bind a variety of different pTyr peptides, but not the non-phosphorylated counterparts (Fig. 3d; Supplementary Fig. 6a and b; Supplementary Tables 3 and 4 for peptide sequences and spot intensities). That the *Legionella* SH2 domains could bind to the majority of the peptides on the arrays suggests they are promiscuous binders (Supplementary Fig. 6a and b). Never-theless, sequence LOGO analysis indicated that LeSH had a proclivity for acidic residues whereas the RavO SH2 domain preferred hydrophobic residues at the +1 position (Supplemen-tary Fig. 6c; Supplementary Tables 5 and 6 for *z*-scores). The specificity of either the LeSH or RavO SH2 domain is, however, distinct from that of a mammalian SH2 domain[29].

Based on the array-binding profiles (Supplementary Fig. 6a and b), we identified a group of pTyr peptides that showed strong binding to either the LeSH or RavO SH2 domain. These peptides were re-synthesized individually with a fluorescein tag at the N-terminus (and a GG- or a 6-aminohexanoic acid spacer) and used to determine in-solution affinities for 13 different *Legionella* SH2 domains (Fig. 3c). Nine exhibited strong affinities ($K_d < 1\ \mu M$) for one or more pTyr peptides. The affinities for these *Legionella* SH2 domains were usually higher than the Src SH2 domain for the same peptides (Fig. 3c). Of note, the *L. drancourtii* LeSH4 and *L. Longbeachae* LUSH SH2 domains bound to the GG-pY-GG and the pY-G dipeptide with affinities substantially surpassed those observed for the Src SH2-derived superbinder[34] (Fig. 3b, c). The remarkable affinities exhibited by the *Legionella* SH2 domains have never been observed on a mammalian SH2 domain, and indeed any naturally occurring SH2 domain characterized to date. The pTyr-superbinding property notwithstanding, the LUSH and LeSH4 SH2 domains bound to the other pTyr peptides with comparable, and in certain cases even lower affinities as to the pY-G dipeptide (Fig. 3c). This indicates that, while the *Legionella* SH2 domains may be optimized for pTyr recognition, they have not evolved an elaborate mechanism to recognize residues flanking the pTyr as do mammalian SH2 domains[30].

**The SH2 domains bind to tyrosine-phosphorylated host proteins**. We next addressed whether the *Legionella* SH2-containing proteins could bind host proteins in a tyrosine phosphorylation-dependent manner. To this end, U937 human macrophage-like cells were treated with the phosphatase inhibitor pervanadate (pv) to enrich phosphorylated proteins prior to preparing cell lysate[48]. Far-western blots using GST-fused proteins showed that the four *Legionella* SH2 domains (LeSH, LUSH, LeSH4, and RavO) tested were all capable of binding the host cell proteins in a pv-dependent manner (Supplementary Fig. 6d). Moreover, the binding pattern of the *Legionella* SH2 domain was virtually indistinguishable from that for the Src SH2 superbinder. In complementary experiments, we employed GST-fused SH2 domains to pull down Tyr-phosphorylated proteins from pv-treated macrophage cells. Of the seven *Legionella* SH2 domains tested, all but one (*Ldu* LeSH1a) showed a robust ability to pull down tyrosine-phosphorylated proteins (Fig. 3e; Supplementary

Fig. 6e). Mutating the conserved Arg-βB5 residue (see Fig. 1a) on LeSH (i.e. in the R46K mutant) or the RavO SH2 domain (R266K) resulted in a substantial loss of binding to tyrosine-phosphorylated proteins (Fig. 3e) and peptides (Supplementary Fig. 7a and b).

Results from the peptide array and in-solution binding assays suggest that the mammalian adaptor protein Shc1 is a potential binding target of the *Legionella* SH2 domains. Indeed, Shc1 was pulled down by the *Legionella* SH2 domains from the U937 macrophages treated with pervanadate (Fig. 3e; Supplementary Fig. 6e). From mass spectroscopy analysis, we also identified valosin-containing protein (VCP, also called p97 or CDC48) as a major target for LeSH. VCP and its cofactor proteins play an essential role in the replication of *L. pneumophila* in mammalian cells[49]. VCP was indeed precipitated by the *Legionella* SH2 domains in a pTyr-dependent manner (Fig. 3e). Binding of VCP was apparently mediated by phosphorylation of the C-terminal Tyr805 residue since the dipeptide pTyr-Gly bound strongly to the *Legionella* SH2 domains (Fig. 3c). This penultimate tyrosine residue in VCP is conserved in some natural host species, although it is as-of-yet unknown whether the site is phosphory-lated in protozoans (Supplementary Fig. 6f). Taken together, our binding assays implicate VCP, Shc1 and numerous other host proteins as potential targets for the *Legionella* SH2-containing effectors during pathogen–host interactions.

**Molecular basis for pTyr recognition by the SH2 domains**. To understand the molecular basis underlying the superbinding characteristics of the *Legionella* SH2 domains, we determined the structures of *L. longbeachae* LeSH and the *L. pneumophila* RavO SH2 domain in complex with tyrosine-phosphorylated peptides by X-ray crystallography (Fig. 4a–c; Supplementary Figs. 8a–d and 9a–e; Supplementary Table 7). We obtained crystal struc-tures of LeSH in complex with three different pTyr ligands, including the phosphotyrosine itself, a 13-mer peptide derived from the pTyr381 site of DnaJ-A1, and another 13-mer peptide corresponding to the pTyr387 site in IL2Rβ. Apart from the pTyr residue, which assumed the same conformation in all three complexes, we were not able to detect the electron density for residues preceding the pTyr in the DnaJ-A1 peptide or any residues flanking the pTyr in the IL2Rβ peptide (Supplementary Fig. 8b and d) despite the high resolution (1.6–1.7 Å) of the complex structures obtained. This suggests that, in contrary to their important role in binding to a mammalian SH2 domain, residues C-terminal to the pTyr may not make a significant contribution in binding to a *Legionella* SH2 domain.

By soaking a 10-residue peptide derived from Shc1 (pTyr317) with the RavO SH2 domain crystal, we obtained the complex structure and found that the peptide was identifiable in three out of the four SH2 domain molecules in an asymmetric unit (Supplementary Fig. 9e). The conformation of the bound Shc1 peptide for the residues between Ser-1 and Val+3 is similar for the three copies in the asymmetric unit (Supplementary Fig. 9f). In both the apo and peptide-bound structures, one RavO SH2 molecule forms a dimer with another in the crystal through a hydrophobic interface (Supplementary Fig. 9d and e).

Both the LeSH and RavO SH2 domains contain the SH2 fold (Fig. 4a–d; Supplementary Fig. 10a). The DALI structural comparison server[50] detected the human SAP SH2 domain (16% sequence identity) as the top structural neighbor for LeSH, whereas the RavO SH2 domain most closely resembles the yeast Spt6 SH2 domain structurally even though the two share only 13% sequence identity (Fig. 4e). Compared to the Src SH2 domain, LeSH is missing two β-strands (βA and βF) but contains a 45-residue insert between the strand βE and the helix αB (Fig. 4d).

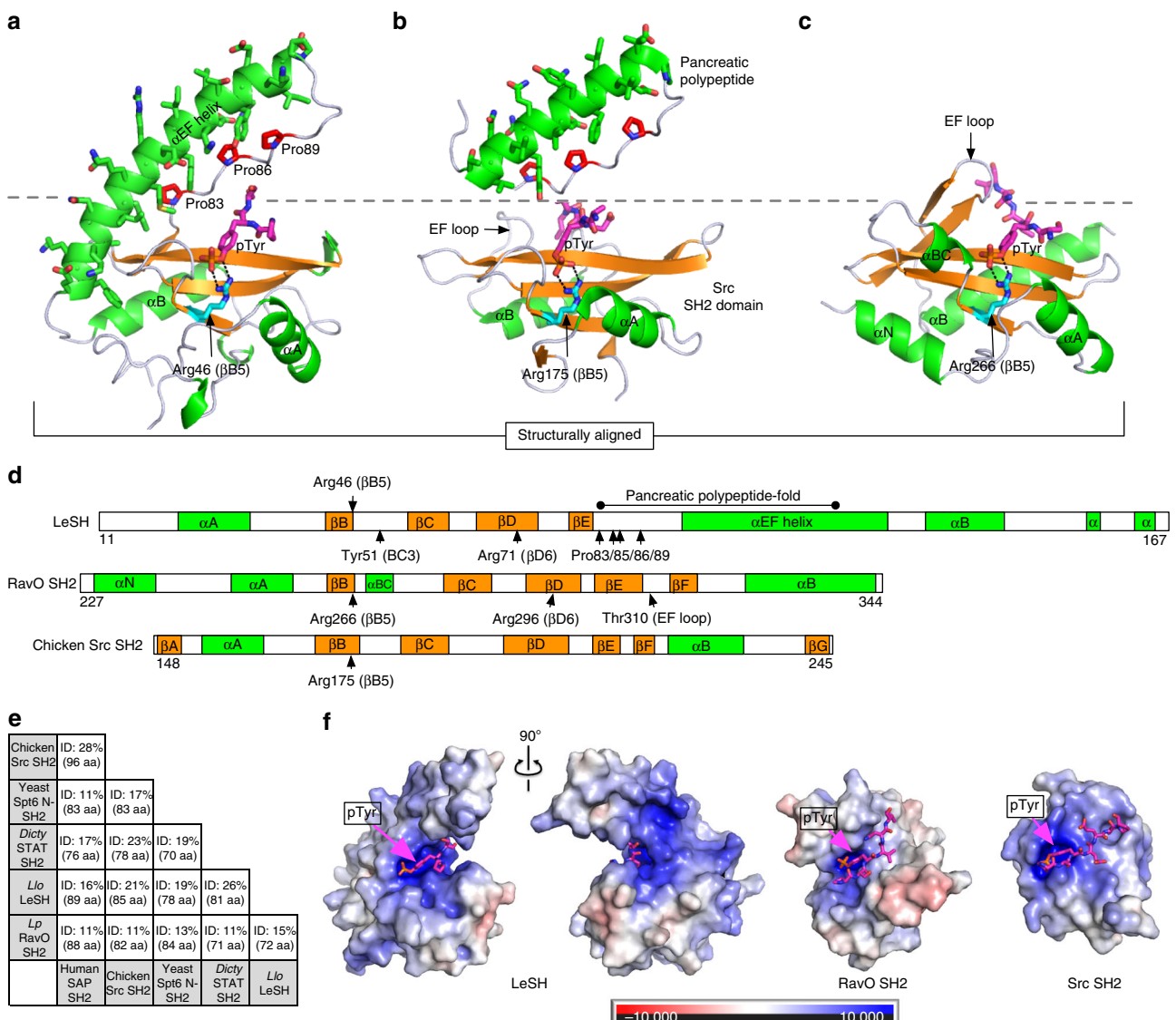

**Fig. 4** Structures of two *Legionella* SH2 domains with bound phosphopeptides. **a** Crystal structure of *L. longbeachae* LeSH in complex with the IL2Rβ pTyr387 peptide. Only three residues (Pro-pTyr-Ser) could be traced in the electron density (see Supplementary Fig. 8d). The three prolines associating with the hydrophobic side of the protruding αEF helix are shown in sticks and colored red. See Supplementary Fig. 8e for amphipathic nature of the helical extension. **b** The Src SH2 domain with the avian pancreatic polypeptide shown on top to mimic the LeSH structure. Pro residues in the pancreatic polypeptide that make contact with the hydrophobic side of the α-helix (αEF) are colored red. **c** Crystal structure of the RavO SH2 domain in complex with the Shc1 pTyr317 peptide. Panels **a** and **c** are structurally aligned to the Src SH2 domain. Helices and β strands in the SH2 domains are shown in green and orange ribbons, respectively. The conserved βB5 arginine is colored cyan. The bound phosphopeptides are in magenta sticks. **d** A comparison of secondary structures for the three SH2 domains. **e** Sequence identity between *Legionella* and selected eukaryotic SH2 domains based on structure-based alignments. SH2 domains from the chicken Src, human SAP, *Dictyostelium* STAT, and yeast Spt6 protein were included for the structure-guided sequence identity calculation. The total number of aligned residues in each pairwise structural comparison was given in the parenthesis. **f** Electrostatic surface potential of the LeSH, RavO, and Src SH2 domains drawn with the same scale. The bound peptides are shown as magenta sticks. The surface is colored in a gradient from red (negative) to blue (positive electronic potential)

This extra sequence in LeSH contains a structural motif called the pancreatic polypeptide (PP)-fold[51], comprising an amphipathic α-helix (named herein as αEF) packed against a polyproline segment via favorable hydrophobic contacts (Supplementary Fig. 8e). Therefore, LeSH may be considered a hybrid of two eukaryotic protein folds – SH2 and PP. Because the PP-fold occupies the same position as the EF loop in a eukaryotic SH2 domain (Fig. 4d), they may play a similar role in ligand binding. Indeed, the SH2 and PP folds together form a clamp-like structure that grasp the pTyr residue (Fig. 4f). An overlay of the LeSH structures in apo and peptide-bound forms identified the PP-fold as the only

region in LeSH that undergoes a significant conformational change upon peptide binding (Supplementary Fig. 8f). Molecular simulation suggested that the motion of the PP-fold was distinct from that of the SH2 fold (Supplementary Fig. 8g). Therefore, it is likely that the pTyr-clamp in LeSH may be able to adjust its conformation to a certain degree to accommodate different peptides. This mechanism of ligand binding may be employed by other *Legionella* SH2 domains of the LeSH subfamily (Supplementary Fig. 1). In contrast, the RavO SH2 domain has a much shorter EF loop compared to LeSH, and employs a distinct mode of ligand recognition as described in more detail below.

**A defined pTyr-binding pocket for high-affinity binding**. Both the LeSH and RavO SH2 domains contain a defined pTyr-binding pocket enriched for positively charged residues (Fig. 4f). Of note, the positive charge on LeSH extends well beyond the pTyr-binding pocket due to the presence of a Lys-rich face in the αEF helix (Supplementary Fig. 8e). The expansive positive electrostatic potential explains the general (i.e., position-independent) preference for acidic residues by LeSH beyond the pTyr (Supplementary Fig. 6c). Similar to a eukaryotic SH2 domain, the LeSH and RavO SH2 domains contain Arg residues at the βB5 and βD6 positions to coordinate pTyr binding (Fig. 5a). A relatively conserved mutation of the βB5-Arg46 residue to a Lys led to marked decreases in binding affinities for Tyr-phosphorylated proteins (Fig. 3e) and pTyr-containing peptides (Supplementary Fig. 7a and b). However, the R46K mutant of LeSH bound the DnaJ-A1 and IL2βR peptides that contain multiple Asp/Glu residues markedly better than to the remaining peptides that contain fewer acidic residues. This indicates that charge–charge interaction plays a pivotal role in the LeSH-peptide ligand interaction. In agreement with this assertion, substitution of the βD6-Arg71 in LeSH by Leu resulted in complete loss of binding to all pTyr peptides examined (Supplementary Fig. 7a and b). It is likely that the βD6 Arg residue plays a more prominent role in pTyr binding

in LeSH than in mammalian SH2 domains as an equivalent mutation in the SAP SH2 domain only led to 12-fold reduction in affinity[52] and a Lys(βD6)Leu mutant of the Src SH2 domain even showed a marked increase in pTyr-binding affinity, due apparently to enhanced hydrophobic interaction[34].

To resolve these seemingly contradictory results, we analyzed the conformation of the pTyr sidechain and the pTyr-binding pocket in different SH2-peptide complexes (Fig. 5b). Depending on which Arg or Lys residues of the SH2 domain are involved in forming the salt bridges with the phosphate moiety of the pTyr, the pTyr-binding pocket may be classified into two groups. In the first group, represented by LeSH and SAP, the phosphate is coordinated by the βB5 and βD6 residues on opposite sides. In the second group to which the Src SH2 domain belong, the phosphate moiety forms salt bridges with the ArgβB5 and another Arg/Lys residue at αA2. In the latter case, the βD6 Arg/Lys residue would be too far removed from the phosphate to form a salt bridge with each other although it may still contribute to pTyr binding by interacting with the phenyl ring (Fig. 5b). Thus, the βD6 residue plays distinct roles in the two groups of SH2 domains (Fig. 5b).

The pTyr-binding pockets in LeSH and the RavO SH2 domain are further defined by the unique structures in the region between

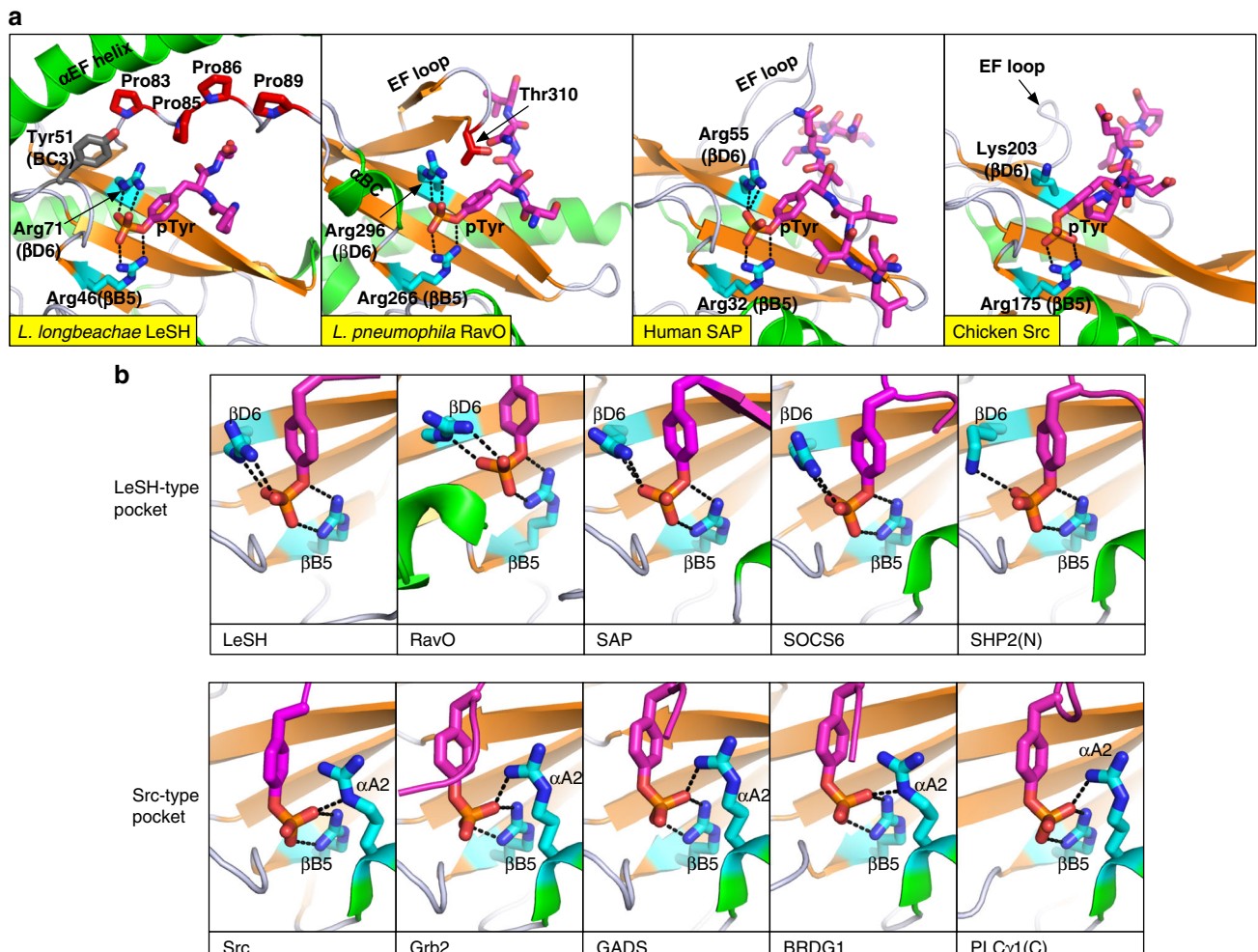

**Fig. 5** Comparison of the pTyr-binding pocket between *Legionella* and animal SH2 domains. **a** The pTyr-binding pockets of the LeSH and RavO SH2 domains. The human SAP and chicken Src SH2 domains are shown for comparison. **b** Classification of pTyr conformation based on structural alignment of the LeSH and RavO SH2 domains with eight eukaryotic SH2 domains. Residues that contribute to salt bridges are shown in cyan sticks. The following high-resolution (resolution ≤2 Å and the crystallographic free-R <0.25) crystal structures were used here: SAP (PDB code 1D4W), SOCS6 (2VIF), SHP2 (N-terminal SH2, 3TKZ), Src (1SHA), Grb2 (1JYR), GADS (1R1Q), BRDG1 (3MAZ), and PLCγ1 (C-terminal SH2, 4K45)

the strand βE and helix αB, which effectively provides a roof for the pocket. Specifically, the Pro85 residue in LeSH and Thr310 in RavO sit atop of the pTyr residue, opposite to the ArgβB5 residue located at the bottom of the pTyr-binding pocket (Fig. 5a; Supplementary Fig. 11a). Besides sculpting the pTyr-binding pocket, the roof residue may play an important role in pTyr binding. Indeed, replacing the roof residue with an Ala (as in the mutants LeSH P85A and RavO T310A) decreased affinities of the mutants by 2.9-fold on average for a panel of pTyr peptides (Supplementary Fig. 7a). This suggests that the pTyr-pocket roof residue in both SH2 domains contribute positively to pTyr binding.

The strong affinities of the LeSH and RavO SH2 domains for the pTyr, as demonstrated using the GGpYGG and pYG minipeptides (Fig. 3c), may be explained by the unique characteristics of the corresponding pTyr-binding pockets. The backbone structure of the N-terminal half of the SH2 domain, the region mediating pTyr binding, is highly conserved between bacterial and eukaryotic SH2 domains (Supplementary Fig. 10b). In contrast, the C-terminal half is highly variable in both sequence and structure between the *Legionella* and Src SH2 domains (Supplementary Fig. 10a and c). Pairwise structure-based sequence alignment detected only two identical residues between the C-terminal half of the Src SH2 and either the LeSH or RavO SH2 domain (Supplementary Fig. 10c). As shown in Supplementary Fig. 10a (middle panel), the EF loop in the Src SH2 domain (important for specificity[28], colored purple), is located on the opposite side of the central β-sheet to the βB5-Arg175 residue (critical for pTyr binding, colored cyan). However, the αEF helix in LeSH (colored purple, equivalent to the EF loop in Src SH2) and βB5-Arg46 are located on the same side (Supplementary Fig. 10a, left panel) of the β-sheet. Similarly, the EF loop of the RavO SH2 domain (colored purple) is located close to the pTyr-binding βB5-Arg266 residue (Supplementary Fig. 10a, right panel). Instead of being part of the specificity pocket, the αEF helix of LeSH and the EF loop of the RavO SH2 domain are an integral part of the pTyr-binding pocket. Accordingly, LeSH makes more extensive contacts with the pTyr compared to a human SH2 domain (Supplementary Fig. 11b).

Furthermore, the LeSH family SH2 domains contain a tyrosine in the BC loop (at BC3, except for Lspi_0399, Supplementary Fig. 1) that may contribute to pTyr binding by stabilizing the pTyr-binding pocket. The crystallographic B-factor distribution pattern provides further supports for Tyr51 in stabilizing the BC loop and potentially minimizing the entropic loss upon ligand binding. The B-factor tends to be higher for loop residues in proteins that reflect structural flexibility, including the BC loop of wild-type SH2 domains (Supplementary Fig. 11c, left panels). The normalized B-factor for Tyr51 in LeSH is similar to the equivalent Val residue in Src SH2-derived superbinder (triple mutant) (Supplementary Fig. 11c). In the RavO SH2 domain, the region equivalent to the BC loop forms a four-residue-long α helix (named herein as αBC), which may contribute to stabilizing the local structure as reflected by the lower B-factor for the BC3 residue (Supplementary Fig. 11a and c). Thus, BC loop stabilization may be a strategy taken by the *Legionella* SH2 domains to achieve a superb affinity for pTyr.

The BC3-Tyr51 residue in LeSH interacts with the hydrophobic face of the αEF helix and with βD6-Arg71, thereby pushing the sidechain of the latter towards the pTyr phenyl ring (Supplementary Fig. 11a). Because the Arg71 residue is sandwiched between Tyr51 of the SH2 domain and pTyr of the ligand for optimal cation–π and charge–charge interactions (Supplementary Fig. 11a), this explains why the LeSH R71L mutant was defective in pTyr binding (Supplementary Fig. 7a and b).

**The *Legionella* SH2 domains lack a specificity pocket.** Eukaryotic SH2 domains typically show specificity for residues C-terminal to pTyr through a specificity pocket[28–30] (Fig. 6a). For example, the Grb2 SH2 domain has specificity for a P + 2 Asn (the second residue C-terminal to the pTyr). The Src or BRDG1 SH2 domain recognizes a hydrophobic residue at the P + 3 or P + 4 position, respectively, via a hydrophobic pocket. However, LeSH lacks a pocket for the accommodation of a C-terminal residue to the pTyr. In fact, only one of the specificity pocket-forming residues in the Src or BRDG1 SH2 domain[30] is found in LeSH (Fig. 6b). Consequently, LeSH features a rather flat surface in the equivalent P + 3/P + 4-binding area (Fig. 6c). This explains the results from the peptide array screening showing minimal sequence selectivity by LeSH (Supplementary Fig. 6c). Furthermore, deletion of either the N- or the C-terminal region from the DnaJ-A1 peptide (DnaJ-A1-pTyr381 ΔN or ΔC) had no apparent impact on binding to LeSH or other *Legionella* SH2 domains (Fig. 3c), suggesting that neither the N- or C-terminal residues contribute to binding. In accordance with this observation, no electron density was detected for residues flanking the pTyr in the DnaJ-A1 peptide in complex with LeSH (Fig. 6c; Supplementary Fig. 8b).

Because both the RavO and Grb2 SH2 domains are capable of binding to the Shc1 pTyr317 peptide with high affinity[53], we used a permutation array[54] of the peptide to distinguish their specificity. As expected, the Grb2 SH2 domain showed absolute selectivity for an Asn at the P + 2 position (Fig. 6d, left panel). The RavO SH2 domain did not show a preference for P + 2 Asn, but instead selected for the hydrophobic residue Val, Ile, and Leu, at the P + 1 position (Fig. 6d, right panel). The observation agreed with the results from the peptide ligand array screen that showed the same residues were preferred at the P + 1 position (Supplementary Fig. 6c). In contrast to the Grb2 SH2 domain that contains a bulky Trp residue (Trp121) at the EF1 position to force the peptide into forming a β-turn structure that is facilitated by the P + 2 Asn residue[53] (Fig. 6e), the equivalent position in RavO SH2 is occupied by Thr (Thr310; Fig. 6f; Supplementary Fig. 9g). Coupled with the movement of the EF loop towards the pTyr-binding pocket, this creates an open binding surface for the C-terminal segment of the peptide ligand similar to what is observed in LeSH. Consequently, the Shc1 pTyr317 peptide was found to bind the RavO SH2 domain in an extended conformation (Fig. 6f). The EF1-Thr310 residue forms part of the roof for the pTyr pocket, but does not contribute to specificity determination. Calculation of the contact area between the Shc1 peptide and the RavO SH2 domain indicates that the pTyr residue makes the most contribution to binding followed distantly by Val + 1 (Fig. 6g). The P + 1 Val may contribute favorably to binding through hydrophobic interactions with Ile340, Phe286, and Ile295 of the RavO SH2 domain (Fig. 6h). Together, our structural and binding data indicate that the *Legionella* SH2 domains have evolved a remarkable ability to bind the pTyr with high affinity, but doing so with minimal selectivity for residues beyond the pTyr.

**Some LeSH homologs are inactive.** Despite having 54% sequence identity with the *L. longbeachae* LeSH, the *L. dumoffii* LeSH1a was inactive in pTyr binding (Fig. 3b, c; Supplementary Fig. 6e). We modeled the pTyr-binding pocket of LeSH1a based on the LeSH structure. We found that the βB7 residue, which forms part of the pTyr-binding pocket, is changed from Ser (Ser48) in LeSH to Asp (Asp44) in LeSH1a (Supplementary Fig. 7c). It is likely that the βB7-Asp residue in the latter serves the role of an intramolecular pseudo-substrate as a pTyr-mimetic, and thereby preventing pTyr binding by LeSH1a. To validate this assertion, we

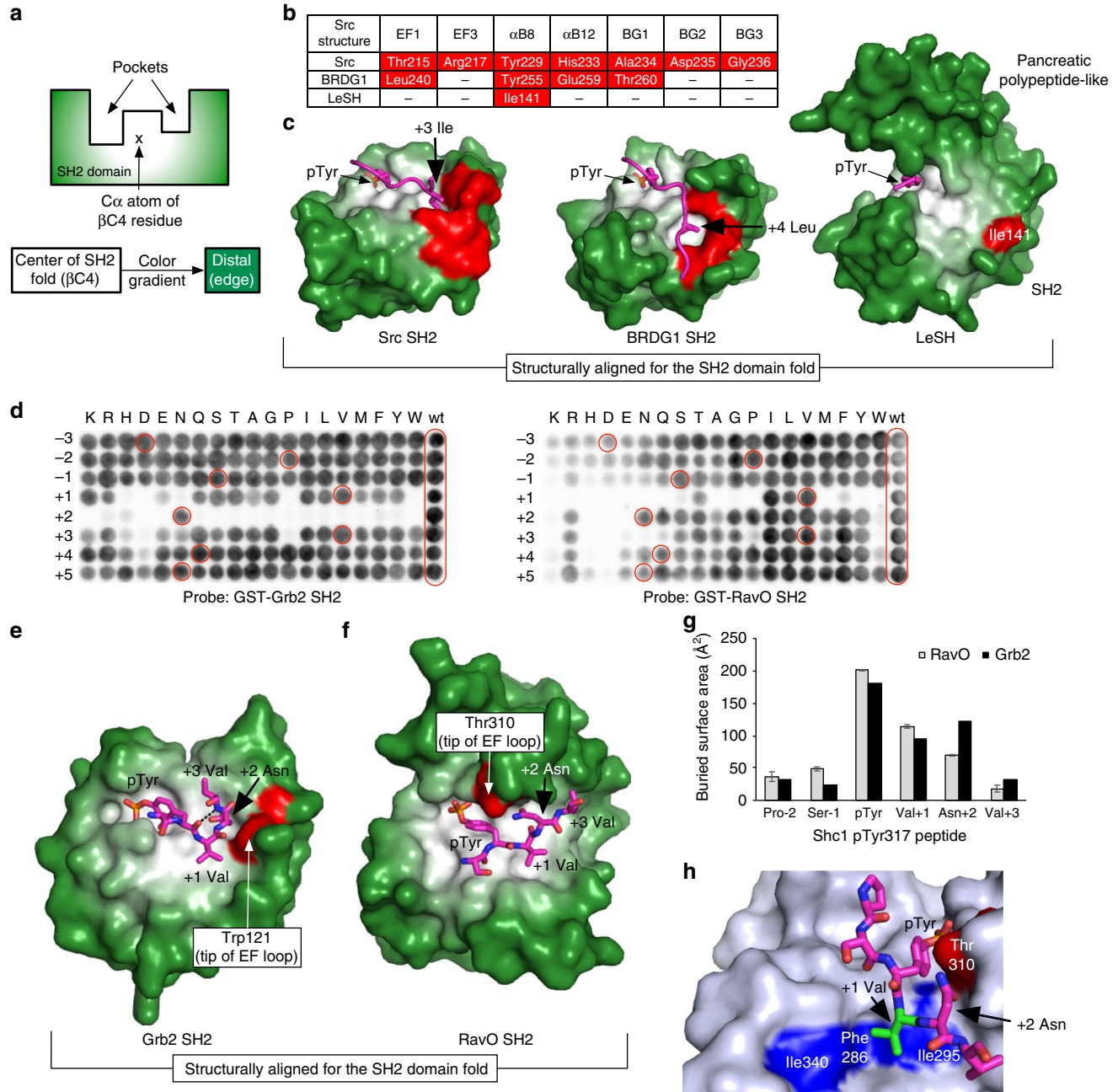

**Fig. 6** The *Legionella* SH2 domains lack a specificity pocket. **a** A color gradient depicting the distance of the ligand-binding pockets on a typical eukaryotic SH2 domain to the center of the globular SH2 domain fold. The Cα atom of the βC4 residue of an SH2 domain was defined as the origin of the color gradient, from white to dark green, to facilitate visualization of the specificity pocket in an SH2 domain. **b** Residues that contribute to forming the inside lining of either the P + 3 or + 4-binding pocket in SH2 domains. The P + 3 (Ile)-binding pocket in the Src SH2 domain involves the EF1, EF3, BG2, and BG3 residues whereas the P + 4 (Leu)-binding pocket in the BRDG1 SH2 domain is formed by the EF1, αB8, αB12, and BG1 residues. A 3D structural alignment of the two SH2 domains with LeSH was used to identify structurally equivalent residues in the latter. The symbol (–) indicates absence of an equivalent residue in the structure. **c** Comparison of the structure of the LeSH-DnaJ-A1 peptide complex with those of the Src and BRDG1 SH2 domains. The bound peptides are pTyr-Glu-Glu-Ile for the Src SH2 domain, and pTyr-Glu-Asn-Val-Leu for the BRDG1 SH2 domain. Only the side chains of the key residues (Ile + 3 for the Src SH2 ligand and Leu + 4 for the BRDG1 SH2 ligand) are shown in magenta sticks. The pocket-forming residues listed in **b** are colored red. **d** A permutation array based on the Shc1 pTyr317 peptide (DPSpYVNVQNL) probed with either the GST-fused Grb2 (left panel) or RavO SH2 domain (right panel). The amino acid at each position between −3 and +5 was replaced by one of 19 natural amino acids (except for Cys). The pTyr residue was fixed. The peptide spots identical to the wild-type (wt) sequence are highlighted in red circles. **e**, **f** Comparison of the LeSH and human Grb2 SH2 domains bound to the same peptide (Shc1 pTyr317). The human Grb2 SH2 domain (**e**) is structurally aligned with the *L. pneumophila* RavO SH2 domain (**f**). The Shc1 peptide Ser-pTyr-Val-Asn-Val is shown in magenta sticks. The residue located at the tip of the EF loop is colored red for each SH2 domain. **g** The buried surface area (in Å²) calculated for each residue of the Shc1 pTyr317 peptide bound to the SH2 domain of RavO or Grb2. The error bars for the area from the RavO complex indicate deviations between the three copies of SH2-peptide complexes in the asymmetric unit (Supplementary Fig 9e and f). **h** The hydrophobic patch on the RavO SH2 domain used for accommodating the +1 position of the ligand peptide

created the LeSH1a-D44S mutant in which the Asp44 residue was replaced by Ser. Indeed, the mutant LeSH1a regained binding, albeit weakly, to several tyrosine-phosphorylated peptides (Supplementary Fig. 7a and d). While it is difficult to predict the physiological function of a pTyr-binding-deficient SH2 domain, LeSH1a does not appear to be an isolated case. The *L. anisa* strain Linanisette encodes three SH2 proteins, namely, LeSH1b, LeSH2a, and LeSH2b (Fig. 2a; Supplementary Table 1). While LeSH2a showed pTyr-binding ability, LeSH1b and LeSH2b appeared inactive. Although the *L. anisa* LeSH1b (Lani_2495) is 55% identical to the *L. longbeachae* LeSH, the former is characterized with a shorter (3-residue) BC loop than the latter (Supplementary Figs. 1 and 2). It is likely that the BC loop in LeSH1b is too short to form a functional pTyr-binding pocket. In support of this possibility, we found that the shortest BC loop in a human SH2 domain contains four residues (Supplementary Fig. 7e). Intriguingly, while LeSH2a (Lani_0711) and LeSH2b (Lani_WP_019234638) are 81% identical to each other, the former bound the GGpYGG peptide with a marked higher affinity than the latter (Fig. 3c). Structural modeling suggests that the Glu80 in LeSH2b may pull the Arg66 (βD6) away from pTyr-binding, thus reducing affinity (Supplementary Fig. 7f). Because the corresponding proteins are capable of translocating to the host cells (Fig. 2b)[13], these pTyr binding-deficient SH2 proteins may play a role in infection independently of phosphotyrosine binding.

## Discussion

The human genome encodes approximately 90 tyrosine kinases and 120 SH2 domains[26,55]. Due to its importance in health and disease, tyrosine phosphorylation has been extensively studied in mammals, but poorly characterized in non-animal species. Moreover, the metazoan-type tyrosine kinases (classified as the Group A tyrosine kinases that include the human tyrosine kinases) have never been identified in fungi or any species more distant than fungi (see Supplementary Fig. 12)[55]. Nevertheless, tyrosine phosphorylation has been identified in non-metazoan species that include natural hosts of *Legionella* such as *Acanthamoeba, Dictyostelium, Hartmannella*, and *Tetrahymena*[23,56,57]. Moreover, some non-metazoan kinases, classified as serine/threonine kinases or tyrosine kinase-like kinases, have been shown to have sporadic tyrosine phosphorylation activities[26,57]. In contrast to tyrosine kinases, SH2 domains and phosphotyrosine phosphatases are found in eukaryotic species that diverged two million years ago (Supplementary Fig. 12). This implies that the tyrosine phosphorylation signalling system is functional in diverse eukaryotic species regardless whether they encode a tyrosine kinase[26].

Besides eukaryotes, serine, threonine, and tyrosine phosphorylation events have been identified in some bacteria, although bacterial tyrosine kinases are evolutionarily unrelated to eukaryotic kinases[58]. Because of the wide distribution of the SH2 domain in eukaryotes, we initially questioned whether there was any prokaryotic SH2-like sequence that might suggest a potential origin for the domain. This inquiry led to the identification of a large family of SH2 domains in *Legionella*, a genus of bacteria that is best known for causing the Legionnaires' disease. Out of the 40 *Legionella* species that contain a putative SH2 domain effector, 28 are known to cause disease in humans. Many *Legionella* effector proteins contain eukaryote-like sequences, and it has been proposed that such sequences have arisen from horizontal gene transfer during pathogen–host interactions[10]. In this regard, it is worth noting that the SH2 domain was originally identified as a region conserved in oncogenic tyrosine kinases in certain avian RNA viruses[59]. Examples of the viral SH2-containing oncoproteins include the tyrosine kinase v-Src, the adaptor protein v-Crk, and the ubiquitin ligase v-Cbl. These viral proteins are nearly identical to their counterparts in the animal host, suggesting that the viral SH2 domains have originated from the host and upon infection, may play a role in hijacking phosphotyrosine signalling in the host cell[60]. Although it is likely that the *Legionella* SH2 domains are originated from eukaryotic species, there are no identifiable eukaryotic homologs for any of the 84 SH2-containing bacterial proteins presumably due to substantial evolution of these proteins after their acquisition by the bacteria.

Besides the large number of *Legionella* SH2 domains uncovered, it is remarkable that the vast majority of the bacterial SH2 domains examined in this study are bona fide pTyr binders, and in certain instances, pTyr superbinders. Our structural analysis of the LeSH and RavO SH2 domains revealed two distinct modes of pTyr binding, suggesting that additional modes of pTyr recognition may still exist for this large family of bacterial SH2 domains. Intriguingly, none of the *Legionella* SH2 domains examined herein exhibited specificity for the sequence motifs identified for mammalian SH2 domains[30]. The lack of specificity for the *Legionella* SH2 domains can be explained by the structure of the LeSH and RavO SH2 domains. While both SH2 domains contain a pTyr-binding pocket, they are devoid of a specificity pocket for binding peptide residues C-terminal to the pTyr. The lack of sequence specificity for the *Legionella* SH2 domain raises a fundamental question about the evolutionary origin of SH2 specificity. It has been proposed that a motif may coevolve with a motif-binding pocket[61]. However, we do not know whether the motif-based recognition plays a role in shaping the SH2-ligand interaction in amoebae or other protozoan host species that do not encode metazoan-like (Group A) tyrosine kinases (which themselves prefer substrates containing specific sequence motifs). Therefore, it is possible that the *Legionella* SH2 domains evolved independently of tyrosine kinases and were therefore not selected for motif recognition to correlate with tyrosine kinase activation and signalling. Since the natural host species of *Legionella* span almost the entire eukaryotic tree of life, it is reasonable to assume that the *Legionella* SH2 domains have evolved to shed the specificity pocket in lieu of promiscuous binding to cope with the diverse target sequences found in distant host species.

We previously created several mammalian SH2 domain mutants, called pTyr superbinders, that bind to tyrosine-phosphorylated peptides with affinities that are orders of magnitude greater than the corresponding natural domains[34]. It is remarkable that at least two of the *Legionella* SH2 domains examined in this study, namely the LUSH and LeSH4 SH2 domains, bound to the pTyr (as in the pYG and GGpYGG mini-peptides) with affinities surpassing that of the Src SH2-derived superbinder. Nevertheless, unlike the Src SH2 superbinder, these bacterial SH2 superbinders did not discriminate one peptide from another based on the pTyr flanking sequence. Therefore, a *Legionella* SH2 domain could potentially bind any Tyr-phosphorylated proteins in the host, thereby making it a true disruptor of phosphotyrosine signaling. However, no significant defect in infection or growth efficiency was observed for *L. longbeachae* mutants in which either or both of the *LeSH* and *LUSH* genes were deleted. *Legionella* secretes hundreds of effector proteins into its host upon infection, so a large degree of redundancy in function is expected of the translocated effectors[13]. The remarkable sequence diversity for the different *Legionella* SH2 domains makes it difficult to exhaust all SH2 domains by sequence-based search; it is therefore possible that additional SH2 domain effectors may exist in a given *Legionella* species that have eluded our identification. Nonetheless, the preponderance of SH2 domains in *Legionella* and the superb pTyr-binding affinities

exhibited by them suggests biological functions for the bacterial SH2-containing effector proteins.

## Methods

**Identification of bacterial SH2 domains by sequence-structure analysis.** Cazalet et al.[11] reported the genome sequence of *L. longbeachae* and annotated the *llo2327* gene as an SH2 domain-containing sequence, which we call LeSH in this paper. We first ran the PSIPRED secondary structure prediction server[62] to confirm that the sequence has the signature secondary structure elements for the SH2 domain fold (at least three β-strands flanked by an α-helix on each side, i.e., [α-helix A]–[three or more β-strands]–[α-helix B], Fig. 1a). Using the *Llo* LeSH as the query, we searched for homologous sequences using NCBI BLAST by limiting the species to bacteria. The candidate homologs from the BLAST search were evaluated first by pairwise alignment between the query (*Llo* LeSH) and each hit sequence, and the sequence is retained if the βB5 arginine (*Llo* LeSH Arg46) is conserved in the hit sequence. The hit sequences were considered as SH2 domains homologous to *Llo* LeSH if the sequence identity is at least 40%. If the sequence identity between the query and the hit sequence is below 40%, secondary structure prediction was performed. A sequence from the BLAST search is considered a putative SH2 domain if (1) the predicted secondary structure shows the α-helix–β-sheet–α-helix SH2-fold signature and (2) the conserved arginine (aligned to *Llo* LeSH βB5-Arg46) is located on the predicted βB strand (the first strand after the αA helix).

The newly identified LeSH homologs were used as queries for a new round of BLAST searches. The same set of criteria (signature secondary structure elements and βB5-Arg conservation) was applied to identify additional bacterial SH2 domain sequences, which were in turn used as search queries in the next round of BLAST searches. The BLAST searches were repeated until no more bacterial SH2 domain homologs that satisfy the criteria were identified. The proteins were grouped according to the pairwise sequence identity as well as the LOG (*Legionella* orthologous group) sequence clustering reported by Burstein et al.[13]. The repetitive BLAST searches starting from the *Llo* LeSH sequence identified the LOG clusters LeSH/LeSH1a/LeSH1b (LOG_02684), LeSH2 (LOG_02859), LeSH3 (LOG_07829), LeSH4 (LOG_04518), and LUSH (LOG_02977).

To identify more bacterial SH2 domains without homology to *Llo* LeSH, we searched for sequences that contain the SH2 domain sequence profile from all bacterial proteins. All non-redundant bacterial protein sequences were extracted from the UniRef database. The profile hidden Markov model for the SH2 domain (PF00017.23) was obtained from Pfam website[35]. The hmmsearch program in the HMMER suite[63] was used to scan the bacterial protein sequences for the SH2 domain. The search identified Lpp3070 (DoSH from *L. pneumophila* Paris) that contains two SH2 domains that satisfy the secondary structure and arginine conservation criteria. Using the DoSH sequence as a query, repeated BLAST searches identified members of DoSH (LOG_00141), RavO (LOG_05208), and LeSH5 (LOG_04812) proteins.

The multiple sequence alignment server MAFFT[64] was used for each homologous group to confirm that the βB5 arginine is conserved in all sequences for the group (e.g. Supplementary Figs. 1 and 3a).

Although we did not limit the range of bacterial species in our sequence searches, we could identify SH2 domains with high confidence only from the order *Legionellales*, except for two proteins (UniProt accessions K2BDF1 and K2EE31) from uncultivated species for which the species were unidentified[65]. We synthesized the two genes, expressed and purified proteins and conducted in-solution peptide binding assay. However, we did not detect binding to tyrosine-phosphorylated peptides, and, therefore, we did not include them in this paper.

**PDB IDs used for structure comparison.** The following PDB coordinate files were used for structure-guided sequence alignment and structure comparison analysis; PDB 1SPS: chicken (identical to Rous sarcoma virus) Src SH2, PDB 1D4W: human SAP SH2, PDB 1UUR: social amoeba *Dictyostelium* transcription factor STAT, PDB 3GXW: yeast transcription factor Spt6, PDB 3MAZ: human BRDG1 SH2, PDB 1JYR: human Grb2 SH2, PDB 2BF9: avian pancreatic polypeptide, PDB 4WZ3: *L. pneumophila* effector protein LubX.

**Sequence alignment.** Structure-guided sequence alignment in Fig. 1a was prepared following the procedures described below. We initially prepared two subsets of alignments. The first alignment consists of seven sequences: *L. longbeachae* LeSH and LUSH SH2; *L. dumoffii* LeSH1a, LeSH2, and LeSH3; *L. drancourtii* LeSH4, and chicken Src SH2. Two of them (LeSH, for which we solved the structure, and Src SH2) were associated with structure coordinates. The second alignment consists of eight sequences: *L. pneumophila* RavO and DoSH C-terminal SH2 domains, *L. waltersii* LeSH5 N- and C-terminal SH2 domains, *L. brunensis* Lbru_2627 N- and C-terminal SH2 domains, yeast Spt6 SH2 and chicken Src SH2. Three of them (RavO SH2, for which we solved the structure, Spt6 SH2 and Src SH2) were associated with structure coordinates. A prototype structure-guided alignment for Fig. 1a was produced by the STRAP[66] program by incorporating all the sequences (along with associated coordinates for the LeSH, RavO, and Src SH2 domains). However, the program did not produce a reliable alignment for the C-terminal region, presumably due to large structural diversity and poor sequence conservation (see Supplementary Fig. 10a and c). To deal with the issue, sequence gaps in

the prototype alignment was manually adjusted by referring to the two previous subsets of alignments mentioned above, where the Src SH2 domain served as the common reference to combine the two subsets of alignments.

The multiple sequence alignments in Supplementary Figs. 1, 3a, 4a and 4d were generated by the MAFFT server[64]. The alignment figures and the associated average distance phylogenetic trees were prepared using Jalview[67]. For the statistics of BC loop length in human SH2 domains (Supplementary Fig. 7e), sequences of the 120 human SH2 domains were aligned using the MAFFT server, and the BC loop region was extracted. The number of domains in Supplementary Fig. 12 was taken from Suga et al.[55]. The time-calibrated phylogenetic tree in Supplementary Fig. 12 was derived from Parfrey et al.[68].

**Expression constructs for protein production in *E. coli*.** The list of primer sequences used for molecular cloning and site-directed mutagenesis is provided in Supplementary Table 8. For protein overexpression in *E. coli* to prepare purified recombinant proteins for biochemical and crystallographic studies, we prepared the *Legionella* genes by gene synthesis (Bio Basic), except for the *L. dumoffii* LUSH, which was cloned from the genomic DNA by PCR. Full-length synthetic genes were prepared for expression of the following proteins: *L. longbeachae* LeSH; *L. dumoffii* LeSH1a, LeSH2, and LUSH; *L. pneumophila* LeSH2; and *L. anisa* LeSH1, LeSH2a, and LeSH2b. Genes corresponding to the following amino acid residue ranges were chosen for expression of the SH2 domains in *E. coli*. *L. longbeachae* LUSH SH2 domain: Ser3-Ser171, *L. drancourtii* LeSH4 SH2 domain: Ser3-Asp170, *L. waltersii* LeSH5 tandem SH2 domains: Asp3-Ala219, *L. pneumophila* DoSH C-terminal SH2 domain: Glu366-Pro546, and *L. pneumophila* RavO SH2 domain: Gly212-Asp406. The hexahistidine (His$_6$)-tagged and His$_6$-GST-tagged constructs were prepared by subcloning the genes into the pETM-11 and pETM-30 vectors[69], respectively. The expression construct for the triple mutant Src SH2 domain (superbinder) was described previously[34].

**Effector translocation assay.** The full-length gene encoding *Legionella* SH2 proteins was subcloned into the pCyaA fusion vector[45]. *L. pneumophila* strains Lp02 and Lp03 expressing the Cya-fusion protein were used for the assay. *L. pneumophila* were grown at an initial OD$_{600}$ of 0.7 with 0.1 mM of isopropyl β-D-1-thiogalactopyranoside (IPTG) until late-exponential phase with large motility. The U937 cells (ATCC CRL-1593.2) were grown in RPMI media (Sigma) supplemented with 10% fetal bovine serum, and were differentiated with 12-tetradecanoyl phorbol 13-acetate for 3 days. The differentiated cells were seeded in 12-well tissue culture plates at a density of $1.0 \times 10^6$ cells/well and challenged with bacteria strains at a multiplicity of infection of 5. After incubation at 37 °C for 1 h, the U937 cells were washed three times with phosphate-buffered saline (PBS), lysed by adding 200 μl lysis solution (50 mM HCl, 0.1% Triton X-100), and incubated on ice for 10 min. Lysed samples were boiled for 5 min, and then neutralized with 10 μl of 1 M NaOH. After addition of 400 μl of cold 95% ethanol and incubation on ice for 5 min, insoluble material in the extracts was removed by centrifuging at 4 °C at $16,000 \times g$ for 5 min. Supernatants were vaporized under vacuum and pellets were resuspended in EIA buffer (Cayman Chemical). The levels of cyclic AMP were measured using Cyclic AMP EIA Kit (Cayman Chemical). The CyaA fusion proteins in bacterial lysate were detected by western blot, using mouse anti-CyaA antibody (Santa Cruz Biotechnology, catalog # sc-13582) diluted 1:5000 and goat anti-mouse IgG-HRP conjugate (Bio-Rad, catalog # 170-6516) diluted 1:3000.

**Protein expression and purification for biochemical assays.** Expression of the His$_6$- or His$_6$-GST-fusion proteins were induced with 0.3 mM IPTG in *Escherichia coli* BL21(DE3) at 18 °C, overnight, except for the Src SH2 domains expressed at 30 °C for 6 h. Bacterial cells were lysed in 20 mM sodium phosphate, 20 mM imidazole, 100 mM NaCl, 1 mg/ml lysozyme, 0.5% Triton X-100, pH 8.0, by incubating the cells on ice for 30 min. The Ni-NTA (Qiagen) nickel affinity chromatography was conducted by following the manufacturer's instruction. Samples eluted from the column were dialyzed against 20 mM Tris-HCl, 150 mM NaCl, at 4 °C for 48 h (supplemented with 0.5 mM Tris(2-carboxyethyl)phosphine for the wild-type Src SH2 domain as the reducing agent).

**Autoubiquitination assay.** The full-length *L. longbeachae* LUSH protein fused with the His$_6$-GST tag was used for the assay. The *L. pneumophila* ubiquitin ligase LubX C-terminal truncation (residue 1–186) was used as the positive control for the in vitro ubiquitination assay. The human E2D2 was used as the ubiquitin-conjugating enzyme E2. The experiment was conducted in 50 mM Tris-HCl, 100 mM NaCl, 10 mM ATP, 10 mM MgCl$_2$, 0.5 mM dithiothreitol, overnight at 25 °C. See Quaile et al.[41] for detailed experimental procedure.

**Peptide array.** The peptides were synthesized on a nitrocellulose membrane on a MultiPep synthesizer (Intavis). Either a Gly residue (Supplementary Fig. 6a and b) or a 6-aminohexanoic acid residue (Fig. 6d) was added at the C-terminus of each peptide as a spacer between the peptide and the membrane. To identify tyrosine-phosphorylated peptides that bind to the LeSH and RavO SH2 domains, we prepared two sets of peptide array membrane pairs, Array-1 and Array-2, that were probed by the GST-LeSH and the GST-RavO SH2 domains (i.e. four separate membranes in total, Supplementary Fig. 6a and b). Each membrane consists of 95

Tyr-phosphorylated peptides and five non-phosphorylated peptides. The first 20 peptides are identical between Array-1 (A1-A20) and Array-2 (E1-E20). For the remaining 80 spots in each array, peptide sequences from known human tyrosine phosphorylation sites were selected in the order of the scoring matrix-assisted ligand identification (SMALI) score for the human Fyn (Array-1) or Grb2 (Array-2) SH2 domain, derived from the SMALI program that predicts binding sites for human SH2 domains[47]. In addition to the first 20 peptides, four more peptides were identical between Array-1 and Array-2. These 24 identical peptides were used for normalizing signal intensities between the two membranes for each probe (Supplementary Fig. 6a and b). The membranes were blocked in 5% non-fat milk in TBS-T (20 mM Tris-HCl, 137 mM NaCl, 0.1% Tween-20, pH 7.6), probed with 2 μg/ml GST-fusion SH2 domain at room temperature, and the bound GST-fusion probe was detected with anti-GST antibody-HRP conjugate (Sigma, catalog # A7340). The Two Sample Logo server[70] were used for sequence analysis of the peptide array results.

**Fluorescence polarization assay**. Peptides were synthesized on a MultiPep synthesizer (Intavis) using the standard Fmoc protocol. The peptides were N-terminally labeled with fluorescein, with either two glycines (Gly–Gly) or 6-aminohexanoic acid (ahx) as a spacer between the peptide and fluorescein. The final peptide concentration was adjusted between 1 and 5 nM. Fluorescein polarization was measured on an EnVision multilabel reader 2103 (PerkinElmer) using the wavelength 480 nm for excitation and 535 nm for emission. Average readings from two replicate experiments were used for binding curve fitting assuming one-site peptide binding (see Supplementary Table 2 for curve fitting statistics).

**Far-western blotting**. The U937 cells were treated with phosphatase inhibitor pervanadate at 100 μM in PBS, pH 7.4, for 20 min at 37 °C to enrich phosphorylated proteins, and lysed in 50 mM Tris-HCl, 1% NP-40, 150 mM NaCl, pH 7.8, on ice, with brief sonication. Twenty micrograms of cell lysate was loaded for each lane of the SDS-PAGE gel for separation followed by transfer to a PVDF membrane. The membrane was blocked in 10% non-fat milk in TBS-T, cut into strips, and each strip was probed with 60 μM GST-fusion SH2 protein. Each strip contains two lysate lanes, one without pervanadate treatment and the other with the treatment. The anti-GST antibody-HRP conjugate was used for detection, and the strips were scanned together to allow for an identical chemiluminescence exposure time.

**GST pull-down assay**. The GST-fusion SH2 proteins eluted from the nickel affinity chromatography were dialyzed against 20 mM Tris-HCl, 150 mM NaCl, pH 7.4, overnight at 4 °C. Eighty picomoles of each GST-fusion SH2 protein was mixed with 200 μg U937 cell lysate for 20 min at room temperature. Glutathione Sepharose beads (GE Healthcare), equilibrated in the lysis buffer, were added for incubation for 20 min at room temperature, followed by GST pulldown and western blot analysis. The following antibodies were used: 4G10 Platinum mouse anti-pTyr antibody (1:3000 dilution; Millipore, catalog # 05-1050), mouse anti-SHC1 antibody (1:4000 dilution; BD Biosciences, catalog # 610878), rabbit anti-VCP antibody (1:10,000 dilution; Abcam, catalog # ab109240), goat anti-mouse IgG-HRP conjugate (1:3000 dilution; Bio-Rad, catalog # 170-6516), and goat anti-rabbit IgG-HRP conjugate (1:3000 dilution; Bio-Rad, catalog # 170-6515). Uncropped scans of the western blots that correspond to Fig. 3e are provided in Supplementary Fig. 13.

**Crystallography of *L. longbeachae* LeSH**. For LeSH crystallization, the His6-tagged LeSH eluted from the nickel affinity column was dialyzed against 20 mM Tris-HCl, pH 7.0, 50 mM NaCl, 1 mM DTT, 0.5 mM EDTA, at 4 °C. The affinity tag was cleaved by TEV protease at room temperature for 2 days. The cleavage reaction product was further purified by Superdex 75 size exclusion chromatography (GE Healthcare) in 20 mM Tris-HCl, 150 mM NaCl, pH 7.0. Crystallization was conducted by either hanging or sitting drop vapor diffusion method by mixing an equal volume of the LeSH sample (with or without a ligand) and the mother liquid. The apo crystal was grown in the following condition: 43 mg/ml LeSH, 0.1 M Tris-HCl, pH 9.0, 20% PEG3000, 0.2 M sodium acetate, at 5 °C. The phosphotyrosine complex crystal was grown in the following condition: 25 mg/ml LeSH, 10 mM pTyr, 20% PEG3000, 0.2 M sodium acetate, 0.1 M Tris-HCl, pH 8.9, at 5 °C. The DnaJ-A1 pTyr381 peptide complex crystal was grown in the following condition: 1.3 mM LeSH, 2.8 mM peptide, 16% PEG3000, 150 mM malic acid, pH 7.0, 0.1 M Tris-HCl, pH 7.8, at room temperature. The IL2Rβ pTyr387 peptide complex crystal was grown in the following condition: 1.5 mM LeSH, 3.1 mM peptide, 12% PEG3000, 0.2 M sodium malonate, at room temperature. Parrafin oil was used as the cryoprotectant for all crystals. Data collection was conducted at 114 K using the RUH3R X-ray generator (Rigaku) and the mar345 detector (MarResearch) with CuKα radiation. Diffraction datasets were processed with iMosflm[71]. A dataset from 741 images with the oscilation angle of 2° (1482° in total) was used for phasing of apo LeSH with the sulfur single-wavelength anomalous diffraction (sulfur SAD) method. The Auto-Rickshaw webserver was used for phasing[72]. The ligand complex structures were solved by molecular replacement. The Coot[73] and the Phenix[74] program suites were used for model building and refinement.

**Crystallography of the *L. pneumophila* RavO SH2 domain**. The RavO SH2 domain apoprotein form was determined from selenomethionine-derivatized RavO [209-406] solved by SAD methods to a resolution of 3.30 Å, and subsequently used to phase crystals of native, full-length RavO that spontaneously degraded into the SH2 domain, using molecular replacement, to a resolution of 1.95 Å. For purification of selenomethione-derivatized RavO[209-409], the region of *lpg1129* coding for this region of the protein was cloned into the p15Tv-LIC vector. For purification of native RavO SH2 domain, full-length *lpg1129* was cloned into the p15Tv-LIC vector, coding for an N-terminal His6 tag and TEV protease cleavage site. After structure determination of the apoprotein RavO SH2 domain, the termini of the expression construct were modified to include the region of *lpg1129* coding for RavO residues 225–344 into the pMCSG53 vector, also coding for an N-terminal His6 tag and TEV protease cleavage site; this construct was utilized for expression of native RavO SH2 domain for crystallization with the Shc1 pTyr317 peptide (acetyl-DPSpYVNVQNL-amide). For native protein purification, *E. coli* BL21(DE3)-RIL or BL21(DE3)-Gold cells were transformed with the appropriate expression plasmid, grown to an OD600 of 0.6 at 37 °C, chilled to 16 °C, and induced overnight with 500 mM isopropyl β-D-thiogalactopyranoside. Selenomethionine-substituted RavO[209-406] was expressed in BL21(DE3)-RIL cells according to the manufacturer's instructions (Shanghai Medicilion). All cells were harvested via centrifugation at 5000 *g* and pellets resuspended in binding buffer (50 mM Hepes (pH 7.5), 100–300 mM NaCl, 10 mM imidazole and 2% glycerol (v/v)), lysed by sonication, and cell debris removed via centrifugation at 30,000 *g*. Cleared lysates were loaded onto a 5 ml Ni-NTA column (QIAGEN) pre-equilibrated with binding buffer, extensively washed with binding buffer containing 30 mM imidazole, and protein was eluted using the above buffer with 250 mM imidazole. The His6-tags were removed by cleavage with TEV protease overnight at 4 °C in dialysis buffer (0.3 M NaCl, 50 mM Hepes (pH 7.5), 5% glycerol, and 0.5 mM tris[2-carboxyethyl]phosphine), followed by binding to Ni-NTA resin and capture of flow through. All RavO crystals were grown with the sitting drop vapor diffusion method. SeMet-substituted RavO[209-406] was crystallized by mixing an equal volume (1 μL) of the protein solution at 55 mg/mL and reservoir solution (1.5 M ammonium sulfate, 0.1 M Bis-Tris pH 6.5) and the crystal was cryoprotected using paratone oil before flash freezing under liquid nitrogen. Apo RavO was crystallized by mixing an equal volume (1 μL) of the protein solution at 90 mg/mL and reservoir solution (0.2 M potassium chloride, 20% (w/v) PEG 5K MME), and the crystal was flash frozen under liquid nitrogen. The RavO SH2 domain + Shc1 phosphopeptide complex was determined by co-crystallization using 0.2 mM pTyr317 peptide, and mixing an equal volume (1 μL) of the protein: peptide solution (protein at 90 mg/mL) and reservoir solution (0.2 M ammonium sulfate, 0.1 M Tris pH 8.5, 25% (w/v) PEG 5K MME), and the crystal was cryoprotected using paratone oil. X-ray diffraction data were collected at the Structural Biology Center, Advanced Photon Source, beamline 19-ID at selenium absorption peak. Data were processed using HKL3000 (ref. [75]). For SAD phasing, Phenix. autosol detected four of the six selenomethionine sites in the asymmetric unit of the RavO[209-406] crystal. Phenix.autobuild and Coot[73] were used for all model building. All *B*-factors were refined and TLS parameterization was included in final rounds of refinement. All geometry was verified using Phenix validation tools and the wwPDB server.

**Structural analysis**. The structure-based sequence identity (Fig. 4e) was derived from the Dali pairwise structure comparison server[50]. PyMOL (Schrödinger) was used for structure drawing. The APBS program[76] were used for molecular surface potential calculation (Fig. 4f). The MultiProt server[77] was used for detecting equivalent positions in the 3D structures (Fig. 6b). The buried surface area for each amino acid was calculated by the PISA server[78] (Fig. 6g; Supplementary Fig. 11b). MODELLER[79] was used for structure modeling in Supplementary Fig. 7, c and f, using the LeSH-phosphotyrosine complex structure as the template. The helical wheel projections server (http://rzlab.ucr.edu/scripts/wheel/wheel.cgi?) was used for Supplementary Fig. 8e. The elNémo server[80] was used for the normal mode analysis (Supplementary Fig. 8g).

## Data availability

Atomic coordinates and structure factors have been deposited in the Protein Data Bank (PDB) with accession numbers 6E8H (LeSH), 6E8I (LeSH-phosphotyrosine complex), 6E8M (LeSH-DnaJ-A1 pTyr381 peptide complex), 6E8K (LeSH-IL2Rβ pTyr387 peptide complex), 6DM3 (RavO SH2 domain), and 6DM4 (RavO SH2 domain-Shc1 pTyr317 peptide complex). All other data are available upon request.

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

## Acknowledgements

We thank Brian H. Shilton, Kristina Jurcic, Sung Ouk Kim, Soon-Duck Ha, Huadong Liu, Chengjun Li, Xing Li (Western University), and Chitong Rao (University of Toronto) for technical assistance. This work was supported in part by grants (to S.S.-C.L.) from the Canadian Institute of Health Research (CIHR), by Natural Sciences and Engineering Research Council of Canada (RGPIN-2014-03641) (to A.W.E.), by NIH grants GM074942 and GM094585 (to A.S. through the Midwest Center for Structural Genomics) and by the Department of Energy, Office of Biological and Environmental Research under contract DE-AC02-06CH11357. S.S.-C.L. held a Canada Research Chair in Functional Genomics and Cellular Proteomics. L.L. held the Shandong Provincial Natural Science Foundation (Grant No. ZR2016CM14), National Natural Science Foundation of China (Grant No. 31770821) and was a recipient of Distinguished Expert of Overseas Tai Shan Scholar.

## Author contributions

T.K. and S.S.-C.L. designed the study; S.S.-C.L. and A.S. supervised the experiments; T.K. conducted sequence analysis with support from L.L.; T.K. and X.L. synthesized peptides; C.V., T.K., X.R. and A.C. contributed to molecular cloning. T.K. and C.V. conducted in-solution peptide-binding assay; T.K. conducted peptide array, GST pulldown, far-Western and LeSH crystallography experiments, and structural bioinformatic analysis; X.R. conducted *Legionella* infection experiments with support from T.K., A.C. and A.WE.; P.J.S. and E.E. conducted RavO SH2 domain crystallization and structure determination; T.S. conducted LUSH ubiquitination assay. T.K. and S.S.-C.L. wrote the paper.

## Additional information

**Competing interests:** The authors declare no competing interests.

