## [Peer Review File · Nature Communications]

Reviewers' comments:

Reviewer #1 (Remarks to the Author):

The pTyr binding SH2 domains are common elements of the eukaryotic cellular machinery and play essential roles in signaling events within the cell. Although the presence of SH2-like domains has been predicted in *Legionella* species through sequence analysis, no experimental data supported this prediction. Here, the authors undertook careful computational and experimental analysis to identify potential SH2-like domains in *Legionella* and related bacteria, showed that many of them not only bind pTyr-containing peptides but display higher affinities than eukaryotic SH2 domains and without distinct selectivity for adjoining amino acids. Their analysis of genome sequences led to the identification of 84 SH2-like domains. These sequences were classified into several clusters based on sequence similarities and gene context. Some of the predicted SH2 domains were part of the predicted effector proteins and the authors showed experimentally that at least one SH2-containing protein is indeed secreted into the macrophage during infection. Next, they expressed 13 predicted SH2 domains and showed that most but not all of them bound pTyr containing peptide(s) with affinity surpassing that of the eukaryotic SH2 domains and with almost no sequence specificity of the peptide beyond the presence of pTyr. Finally, they determined crystal structures of two of these SH2 domains alone and bound to various pTyr-containing peptides. The analysis of 3D structures provided explanation for the promiscuity/lack of specificity and as well as revealed molecular reasons for the lack of pTyr binding by some of the SH2-like domains. The manuscript presents a wealth of information on characterization of these SH2 domains gained from combined application of theoretical and experimental approaches and expands the presence of pTyr binding domains to include also bacteria. This is an important contribution to the field. The findings presented in this manuscript led, of course, to new questions, the most curious being the role of the 'inactive' SH2 domains, the functional advantage for the pathogen from high affinity binding and the lack of selectivity. I hope that the authors will address these questions in their future research.

The manuscript is well written and I only have a few minor comments.

p. 15, bottom: there is an interesting description of the two binding modes of pTyr in the binding sites of SH2 domains, which refers to supplementary Fig. 6c. This figure should be included in the main text as it is important to understand the interesting observation described here.

p. 10, l. 13: please specify what is ahx.

p. 16, l. 7: replace 'sculpturing' by 'sculpting'

p. 16, l. 16: I suggest to replace 'curiously' by 'however' or 'on the other hand'.

Fig. 2a: there are only 71 domains shown graphically here vs 84 indicated in the text and Table S1. What is missing?

Reviewer #2 (Remarks to the Author):

The paper by Kaneko et al. describes the identification of a new family of SH2 domain-containing proteins from *Legionella* bacteria. Using purified proteins, the authors have carried out a thorough biochemical and structural characterization of several of the proteins. They show that the SH2 domains are functional, and some of the bacterial proteins can bind phosphotyrosine alone, independent of the surrounding sequence. Their crystallography analysis shows that the proteins lack the specificity pocket normally seen in eukaryotic SH2 domains, providing a structural explanation for the lack of sequence specificity. Presumably, these proteins play a role in the interactions between the bacteria and their host cells. This is a well-designed and executed paper that will be of interest to investigators studying the evolution of pTyr-based signal transduction.

A few minor suggestions:

1. Page 13, first paragraph: the lack of electron density per se does not necessarily show that

residues C-terminal to the pTyr are not involved in binding. On the other hand, the authors' binding data is persuasive.

2. Page 21, first paragraph of Discussion: the statement that Group A TKs have never been identified in species "phylogenetically distant from humans" is confusing. Group A TKs have been identified in several non-metazoan species, including choanoflagellates, filastereans, and ichthyosporeans. I suggest the authors specifically say, "...never been identified in fungi or any species more distant..."

3. I would suggest moving some of the structural discussion into Supplemental Information. The details are important and thoroughly described, but non-structural biologists may miss some of the main message.

Reviewer #3 (Remarks to the Author):

The manuscript by Kaneko et al describes the characterization of the Src homology 2 (SH2) domains present in different predicted effector proteins in the *Legionella* genus. The authors identified a list of predicted effectors which contain this domain and divided these predicted effectors into 8 different SH2 clusters. To address the question if these SH2 domains are functional, they demonstrated that these predicted effectors bind with high affinity and selectivity to phosphotyrosine (pTyr)-containing peptides as well as identified possible host proteins which might be bound by two of these predicted effectors. Additionally, the structure of two of these SH2 domains was determined in complex with tyrosine-phosphorylated peptides, which was followed by in-depth structure function analysis of these domains. The manuscript addresses an important question regarding the function of eukaryotic domains present in bacteria. However, the aspects related to *Legionella* biology and secreted effector proteins should be strengthened prior to publication.

Major comments:

1. It is not clear how the bacterial proteins that contain the SH2 domains were divided into clusters leading to LeSH1a, LeSH1b, LeSH2, LeSH3, LeSH4, LeSH5, LUSH, RavO and DoSH. According to Supplementary Table 1 some of the LOGs published by Burstein et al. were divided and some were merged, but the basis for this new clustering is not clear.
2. A major concern is the lack of translocation analysis for the SH2 containing predicted effector proteins. Fig. 2A presents the translocation analysis of RavO (which was validated before as an effector protein) and LUSH. However, nothing is known about the translocation of the other predicted effectors analyzed. The authors should examine at least one predicted effector from each of the clusters they generated in order to show that these proteins are indeed translocated into host cells during infection and might perform the function expected from an SH2 domain containing effectors. This point becomes even more relevant due to the results presented in the manuscript showing that few of the SH2 domains are inactive, maybe the predicted effectors containing these domains are not translocated.
3. The data presented in Fig. 2B lacks statistical analysis. The bars showing the levels of cAMP contain no error-bars and the data was not examined for its statistical significance (t-test).
4. It is very hard to interpret the relevance of the results presented regarding the binding of host proteins using the SH2 domains. From one hand, the authors show that the SH2 domains lack a specificity pocket and concluded that the *Legionella* SH2 domains are non-selective pTyr binders. On the other hand, the binding of host proteins by the *Legionella* SH2 domain containing proteins was performed not in the contexts of infected cells. Therefore, it might be that the host proteins identified are simply the most abundant tyrosine-phosphorylated proteins present in U937 cells.

The authors should include a control that will indicate that the host proteins identified are not abundant in U937 cells or show that these proteins are relevant to Legionella infection of host cells.

Minor comments

1. In the manuscript that are several places where the text is written red, and bacterial names are not in italics, this should be fixed.

Manuscript NCOMMS-18-05607

Detailed responses to reviewers

Reviewer's comments quoted verbatim in italics

Reviewer #1:

The pTyr binding SH2 domains are common elements of the eukaryotic cellular machinery and play essential roles in signaling events within the cell. Although the presence of SH2-like domains has been predicted in Legionella species through sequence analysis, no experimental data supported this prediction. Here, the authors undertook careful computational and experimental analysis to identify potential SH2-like domains in Legionella and related bacteria, showed that many of them not only bind pTyr-containing peptides but display higher affinities than eukaryotic SH2 domains and without distinct selectivity for adjoining amino acids. Their analysis of genome sequences led to the identification of 84 SH2-like domains. These sequences were classified into several clusters based on sequence similarities and gene context. Some of the predicted SH2 domains were part of the predicted effector proteins and the authors showed experimentally that at least one SH2-containing protein is indeed secreted into the macrophage during infection. Next, they expressed 13 predicted SH2 domains and showed that most but not all of them bound pTyr containing peptide(s) with affinity surpassing that of the eukaryotic SH2 domains and with almost no sequence specificity of the peptide beyond the presence of pTyr. Finally, they determined crystal structures of two of these SH2 domains alone and bound to various pTyr-containing peptides. The analysis of 3D structures provided explanation for the promiscuity/lack of specificity and as well as revealed molecular reasons for the lack of pTyr binding by some of the SH2-like domains. The manuscript presents a wealth of information on characterization of these SH2 domains gained from combined application of theoretical and experimental approaches and expands the presence of pTyr binding domains to include also bacteria. This is an important contribution to the field.

The findings presented in this manuscript led, of course, to new questions, the most curious being the role of the 'inactive' SH2 domains, the functional advantage for the pathogen from high affinity binding and the lack of selectivity. I hope that the authors will address these questions in their future research.

The manuscript is well written and I only have a few minor comments.

p. 15, bottom: there is an interesting description of the two binding modes of pTyr in the binding sites of SH2 domains, which refers to supplementary Fig. 6c. This figure should be included in the main text as it is important to understand the interesting observation described here.

Response: The original Supplementary Fig. 6c is now moved to the new Fig. 5b.

p. 10, l. 13: please specify what is ahx.

Response: Ahx, 6-aminohexanoic acid, is now specified in both the legend to Fig. 3c and in the Methods section.

p. 16, l. 7: replace 'sculpturing' by 'sculpting'

Response: We fixed this typo.

p. 16, l. 16: I suggest to replace 'curiously' by 'however' or 'on the other hand'.

Response: We made the suggested change where applicable.

Fig. 2a: there are only 71 domains shown graphically here vs 84 indicated in the text and Table S1. What is missing?

Response: Of the 84 SH2 proteins presented in Fig. 2a, 71 are coloured according to the effector probability scores reported in Burstein et al. (Ref. 15). We identified 13 additional proteins from the genomic databases and they are coloured grey (labelled "N/A") in the Fig. 2a since these proteins are not listed in Ref. 15 and the corresponding effector probability scores are not available.

Reviewer #2:

The paper by Kaneko et al. describes the identification of a new family of SH2 domain-containing proteins from Legionella bacteria. Using purified proteins, the authors have carried out a thorough biochemical and structural characterization of several of the proteins. They show that the SH2 domains are functional, and some of the bacterial proteins can bind phosphotyrosine alone, independent of the surrounding sequence. Their crystallography analysis shows that the proteins lack the specificity pocket normally seen in eukaryotic SH2 domains, providing a structural explanation for the lack of sequence specificity. Presumably, these proteins play a role in the interactions between the bacteria and their host cells. This is a well-designed and executed paper that will be of interest to investigators studying the evolution of pTyr-based signal transduction.

A few minor suggestions:

1. Page 13, first paragraph: the lack of electron density per se does not necessarily show that residues C-terminal to the pTyr are not involved in binding. On the other hand, the authors' binding data is persuasive.

Response: We agree with the reviewer's assertion and have toned down the link between electron density in the crystal structure and peptide binding where appropriate.

2. Page 21, first paragraph of Discussion: the statement that Group A TKs have never been identified in species "phylogenetically distant from humans" is confusing. Group A TKs have been identified in several non-metazoan species, including choanoflagellates, filastereans, and ichthyosporeans. I suggest the authors specifically say, "...never been identified in fungi or any species more distant..."

Response: We made the suggested change in the revision.

3. I would suggest moving some of the structural discussion into Supplemental Information. The details are important and thoroughly described, but non-structural biologists may miss some of the main message.

Response: We made the suggested change where appropriate.

Reviewer #3:

The manuscript by Kaneko et al describes the characterization of the Src homology 2 (SH2) domains present in different predicted effector proteins in the Legionella genus. The authors identified a list of predicted effectors which contain this domain and divided these predicted effectors into 8 different SH2 clusters. To address the question if these SH2 domains are functional, they demonstrated that these predicted effectors bind with high affinity and selectivity to phosphotyrosine (pTyr)-containing peptides as well as identified possible host proteins which might be bound by two of these predicted effectors. Additionally, the structure of two of these SH2 domains was determined in complex with tyrosine-phosphorylated peptides, which was followed by in-depth structure function analysis of these domains. The manuscript addresses an important question regarding the function of eukaryotic domains present in bacteria. However, the aspects related to Legionella biology and secreted effector proteins should be strengthened prior to publication.

Major comments:

1. *It is not clear how the bacterial proteins that contain the SH2 domains were divided into clusters leading to LeSH1a, LeSH1b, LeSH2, LeSH3, LeSH4, LeSH5, LUSH, RavO and DoSH. According to Supplementary Table 1 some of the LOGs published by Burstein et al. were divided and some were merged, but the basis for this new clustering is not clear.*

Response: The *Legionella* SH2 domains were clustered by sequence identity and structural features. Our LeSH family classification scheme followed essentially the LOG numbers defined by the Burstein et al. (Supplementary Table 1). There are only two exceptions, Lwal_1101 and Lche_0546. Lwal_1101 was classified as LeSH2, instead of LeSH (LOG_02684), based on sequence alignment and clustering (Supplementary Fig. 1 and 2). Lche_0545 (DoSH1) and Lche_0546 are located side by side on the *L. cherrii* genome, and Lche_0546 is a shorter paralog of Lche_0545 (DoSH1, LOG_00141), and thus we believe that it is reasonable to place Lche_0546 in the DoSH group.

We divided LOG_02684 into three groups (LeSH, and the inactive LeSH1a, LeSH1b), which are distinctively clustered based on sequence identity (Supplementary Fig 2), and show different sequence features (eg., Asp at β B7 in LeSH1a, a short BC loop in LeSH1b, as described in the main text and structure modelling in Supplementary Fig. 7, c-f).

2. A major concern is the lack of translocation analysis for the SH2 containing predicted effector proteins. Fig. 2A presents the translocation analysis of RavO (which was validated before as an effector protein) and LUSH. However, nothing is known about the translocation of the other predicted effectors analyzed. The authors should examine at least one predicted effector from each of the clusters they generated in order to show that these proteins are indeed translocated into host cells during infection and might perform the function expected from an SH2 domain containing effectors. This point becomes even more relevant due to the results presented in the manuscript showing that few of the SH2 domains are inactive, maybe the predicted effectors containing these domains are not translocated.

Response: To address the reviewer's concern, we expanded the translocation assay from 2 to 10 SH2 proteins. As shown in the new Fig. 2b, besides RavO, 7 of the 9 predicted effector proteins were found indeed translocated to the U937-derived macrophage upon *L. pneumophila* infection in a Dot/Icm T4SS system-dependent manner. The translocated effector proteins included *L. dumoffii* LeSH1a and *L. anisa* LeSH1b, which we have shown to be deficient in pTyr-binding. Although it remains to be determined what the functions are for the effector proteins with a defective SH2 domain, the ability of pTyr-binding does not appear to be a prerequisite for effector translocation.

3. The data presented in Fig. 2B lacks statistical analysis. The bars showing the levels of cAMP contain no error-bars and the data was not examined for its statistical significance (t-test).

Response: The new Supplementary Fig. 5 contains detailed data for expression level of the Cya-fusion proteins and p-values that indicate statistical significance of

translocation of the Cya-fusion proteins in comparison to the Cya itself (labelled "no fusion").

4. It is very hard to interpret the relevance of the results presented regarding the binding of host proteins using the SH2 domains. From one hand, the authors show that the SH2 domains lack a specificity pocket and concluded that the Legionella SH2 domains are non-selective pTyr binders. On the other hand, the binding of host proteins by the Legionella SH2 domain containing proteins was performed not in the contexts of infected cells. Therefore, it might be that the host proteins identified are simply the most abundant tyrosine-phosphorylated proteins present in U937 cells. The authors should include a control that will indicate that the host proteins identified are not abundant in U937 cells or show that these proteins are relevant to Legionella infection of host cells.

Response: The data shown in Fig. 3e and Supplementary Fig. 6e were from GST-SH2 pulldown of U937-derived macrophages- a popular cell model for *Legionella* infection. The main conclusion we drew from these experiments is that the *Legionella* SH2 domains are capable of binding to numerous phosphoproteins in the host cells, including Shc1 and VCP, which apparently were not the most abundant Tyr phosphorylated proteins based on the anti-pTyr Western blot (Fig. 3e, upper panel). While it'd be interesting to find out if SH2 binding to Shc1 and/or VCP is relevant to *Legionella* infection of the macrophage, this would involve a large amount of additional experiments which we believe are outside the scope of this work.

Minor comments

1. In the manuscript that are several places where the text is written red, and bacterial names are not in italics, this should be fixed.

Response: These have been fixed in the revised manuscript.